# Maximizing the Impact and ROI of Leadership Development: A Theory- and Evidence-Informed Framework

**DOI:** 10.3390/bs14100955

**Published:** 2024-10-16

**Authors:** Jaason M. Geerts

**Affiliations:** 1Research and Leadership Development, Canadian College of Health Leaders, Ottawa, ON K1S 1V7, Canada; jgeerts3@uottawa.ca; Tel.: +1-(613)-235-7218; 2Cambridge Judge Business School, University of Cambridge, Cambridge CB2 1AG, UK; 3Telfer School of Management, University of Ottawa, Ottawa, ON K1N 6N5, Canada

**Keywords:** leadership development, leader development, evidence-based program design, impact, training transfer, return on investment

## Abstract

Globally, organizations invest an estimated USD 60 billion annually in leadership development; however, the workplace application of learning is typically low, and many programs underperform or fail, resulting in wasted time and money and potential harm. This article presents a novel theory- and evidence-informed framework to maximize the outcomes and return on investment (ROI) of leadership development programs. The foundation of the framework derives from four separate literature reviews: three systematic reviews on leadership development, including the only two to isolate gold-standard elements of effective design, delivery, and evaluation, and one on “training transfer”. Informed by innovative principles of leadership development and unique theoretical models and frameworks, this framework consists of 65 evidence-informed strategies that can be applied as a foundation (9), and before (23), during (17), at the conclusion of (11), and sometime after (5), programs, to maximize impact and ROI. Implications for practice and further research are also presented. Given the stakes, there is an urgent need for evidence and tools to maximize the impact and ROI of leadership development. This novel framework provides robust theory- and evidence-informed guidance for governments, policymakers, and those funding, designing, delivering, and supporting development.

## 1. Introduction

Internationally, an estimated USD 60 billion [1] is spent on leadership development each year, which is roughly a third of the annual funding allocated to cancer treatment [2]. This sum is rationalized by the potential impact of effective leadership on organizations and outcomes [3,4] and by confidence in the importance and utility of professional development [5,6,7,8]. Supporting the latter is evidence that leadership is not purely an innate quality with which one is born, but rather that leader performance is largely attributable to acquired experiences and training [9,10]. Consequently, many organizations consider leadership development an essential cost and a source of competitive advantage [7,11,12,13], as well as a key retention strategy [14,15,16,17,18]. Accordingly, the proliferation of leadership programs offered by organizations internally and those delivered by external providers, including business schools, private corporations, and individual consultants, continues to rise [19]. Unsurprisingly, the substantial time and money dedicated to development carry with them expectations of a return on investment (ROI) and of demonstrable outcomes at the individual level and beyond [5,12,20,21,22,23,24]. This signals a critical need for evidence-informed guidance.

### 1.1. Evidence of Leadership Development Outcomes

The abundance of leadership training corresponds with evidence that interventions can facilitate improved participant outcomes, such as increased confidence, knowledge, skills, capabilities, self-efficacy, engagement, job satisfaction, and performance [25,26,27]. Leadership programs have also been correlated with organizational-level outcomes, such as decreased absenteeism, increased staff retention, engagement, motivation, and sense of shared purpose, and increased organizational performance [25,26,27]. In healthcare settings, this includes superior financial and clinical performance, including improved patient outcomes and patient safety [25,26,27]. These findings echo empirical and meta-analytic results that development programs can facilitate the application of learning to the workplace, commonly called “training transfer” [19,28,29,30,31,32,33,34,35], terms used synonymously here. This application and the achievement of desired outcomes at various levels are indications of the ultimate goal of, and the currency of success in, leadership development, which is increased capacity and outstanding practice [19,36,37]. Finally, there is also evidence that interventions can demonstrate financial ROI [38].

### 1.2. Problem #1: Conflicting Results and Consequences

Despite this evidence, extensive global investment, and typically favorable program evaluations [34,39,40], training transfer, and its corresponding outcomes, are not automatic or certain [7,19,41,42], and the empirical evidence of program effectiveness is conflicting [12,27]. For example, many studies and meta-analyses of leadership interventions report significant, positive results and effect sizes [19,35,39,43]. Others, however, suggest the rate of program participants applying their learning to work is minimal [12,44,45,46,47,48], with as few as five per cent doing so successfully [49]. Furthermore, Avolio et al. (2010) indicate that the organizational ROI of leadership interventions can range from positive benefits worth USD 5,811,600 to losses of USD 460,588 [50]. Additionally concerning is that some programs allegedly fail altogether [51,52], including ten per cent of interventions in a combined data set of 72 studies from two separate systematic literature reviews of leadership development [53]. The frequent inefficacy of these programs has caused many to question the true yield of leadership interventions [12,20,21,54,55] and others to suggest that much of the investment has been wasted [23,56,57,58] and even harmful [23,58]. Poor results create a five-fold loss: wasted funds, the opportunity cost of participants’ time off work, the discouragement of participants, organizational reluctance to invest in further development, and, in certain sectors, the potential to jeopardize people’s well-being [7,59,60,61,62]. Aligned with these results, Beer et al. (2016) concludes that a “great training robbery” is taking place [63].

### 1.3. An Optimizing System

Many studies of unsuccessful interventions attribute their failure to issues related to participants lacking support, change-averse organizational cultures, and the absence of an “Optimizing System”. The latter includes a set of enabling factors designed to maximize program impact and ROI by facilitating the workplace application of learning and the achievement of desired outcomes [7,56,64,65]. These factors relate to the programs (design, delivery, personnel, evaluation, and direct application strategies) and to organizational culture and support. This concept builds on previous terms, including “transfer system” [66], “organizational climate” [67,68], and “training climate” [69]. Although the scholarly literature focuses mainly on the design and delivery of individual interventions, if participants’ organizational cultures are not conducive to change, even outstanding development programs based on gold-standard evidence will underperform or fail [7,63,70,71]. Brinkerhoff and Gill explain, “the workplace can untrain people far more efficiently than even the best training department can train people.” ([72], p. 9) Some argue that application strategies before and after interventions are far more impactful on outcomes than the actual program content and activities [7,48]. Effective design and application strategies can also influence the degree to which individual performance improvements extend beyond participants to organizational levels and become integrated systematically [73].

### 1.4. Problem #2: Research Gaps, Uncertain Quality of Evidence, Dated Models, and Potential Consequences

An additional challenge is the widespread uncertainty regarding which program designs and application strategies individually and collectively are linked empirically to outcomes [40,49,57,63,74,75]. The scholarly knowledge base is predominantly anecdotal, seldom involving objective data [27,53,76,77], which obscures what works best, why, and with what reliability [38], and can dangerously perpetuate ineffective practices [40,53]. Similarly, much of the field’s theoretical foundation, including seminal models and frameworks, is dated, and despite a desire for evidence-based training [78,79,80,81] and available research on the workplace application of learning [7,12,19,44,81,82], there is not yet a comprehensive, theory- and evidence-informed framework of effective approaches to design, delivery, and evaluation and direct application strategies to maximize the impact and ROI of leadership development [12,32]. Designing interventions without a credible theoretical and empirical foundation exposes them and their people to substandard or damaging outcomes [40,51,52,58,78,83]. The risk and fallout of deficient interventions for organizations heighten given that increasingly, budgets are strained, priorities are competing, and capacity is limited [20,84,85].

### 1.5. Purpose

The purpose of this article is to describe a novel theory- and evidence-based framework for maximizing the impact and ROI of leadership development. The framework, called the “Optimizing System”, is grounded in a robust theoretical and empirical foundation, as described in a forthcoming companion article, and is the only published framework of its kind. It was created to support outstanding leadership by providing credible guidance for those designing, revising, delivering, and sponsoring interventions to maximize their quality, impact, and ROI [7,11,69,74,81,82,83,86,87].

### 1.6. Definitions

Defining what we mean by “leadership”, “leadership development”, “integration”, etc., matters because the terms are used so commonly, yet inconsistently, and there is no collective set on which scholars and practitioners generally agree [23]. And yet, how one understands leadership, as distinct from “management” and “power”, for example, has significant implications for how one approaches research, practice, and development program design, delivery, and evaluation [53,88]. Similarly, the predominant term for the enterprise, “leadership development”, nearly always refers to interventions *about* leadership for individuals (thus, *leader* development), often with minimal or no accountability-bound expectations to involve their colleagues. This situation rarely yields remarkable results [7,48], further signaling the need for reliable research and guidance to maximize impact and ROI [23].

To clarify the nomenclature and theoretical underpinnings of the framework [88], the following definitions related to (a) leadership, management, and power; (b) development; and (c) leadership integration are presented.

#### 1.6.1. Leadership, Management, and Power

*Leadership* is the process of leaders and team members collaborating meaningfully and responsibly to realize a shared purpose and vision (True North) [89]. The opposite extreme is power.

*Power* is the capacity to control others without their consent, including through force. In the middle is management.

*Management* is the process of exercising authority to ensure others produce predictable, effective, and efficient results. This typically, but does not automatically, involves positional leaders (supervisors or managers) establishing targets, processes, checkpoints, consequences, and rewards, as well as holding people accountable to these targets. At the team, departmental, and beyond levels, this involves a systematic approach and coordination.

A *leader* is anyone who takes responsibility or is ultimately accountable for the process of realizing a shared purpose and vision (True North) in a given situation [89].

A *team member* is anyone who collaborates meaningfully and responsibly in the process of inspiring others to realize a shared purpose and vision (True North) [89].

*Beneficiaries* refer here to those whom the organization serves, including users, clients, or patients, etc.

#### 1.6.2. Development

*Leader development* is the intentional process of striving to enhance individuals’ leadership knowledge, capabilities, capacity, and performance. This is typically approached through educational, developmental, and/or training programs; experiences; resources; and supports, which can be formal and/or informal and include structured, on-the-job learning, as well as various talent optimization functions [90].

*Leadership development* is the intentional process of striving to enhance collective leadership knowledge, capabilities, capacity, and performance in groups, such as teams, organizations, and communities. This enterprise necessarily includes non-positional leaders and normally addresses the organizational culture required for effective leadership.

*Developmental activities* refer to structured formal and informal experiences that are intended to facilitate development through enhanced knowledge, capabilities, capacity, and performance [71].

#### 1.6.3. Integration

*Talent optimization/management* refers to the structured collection of formal human resources (HR) and organizational development (OD) strategies, requirements, and resources available to staff throughout their tenure intended to maximize their engagement and performance. These include mandatory functions, such as annual performance reviews, and optional ones, such as voluntary training.

*Organizational culture* refers to the explicit and implicit priorities, assumptions, expectations, values, norms, practices, language, and symbols in an organization that convey meaning and influence behavior [91].

*Affected parties* (otherwise, “stakeholders”) refer to those who are likely to be significantly involved in, or impacted by, a given situation.

*True North* elements refer to the collection of an organization’s ultimate raison d’être: its purpose, vision, mission, and values toward which the strategic plan and all other organizational functions should be directed. This also includes organizational standards of excellence.

*A Leadership Organization* (ALO) is a learning organization [92] in which leadership is an expectation of all staff, who are supported by specialized training, corresponding talent optimization, and a conducive organizational culture.

### 1.7. Theoretical Considerations

The Optimizing System framework is informed by the theoretical and empirical foundation described in Geerts (2024) [88], which draws on the wisdom of prominent adult educational theorists, including Knowles [93], Dale [94], Freire [95], Kirkpatrick [96], Kolb [97], and Bloom [98]. The Geerts article presents unique models and frameworks, including progressive levels of mastery, an outcomes-based design, levels of program outcomes, categories of development activities, a program evaluation framework, talent optimization functions, and the concept of A Leadership Organization (ALO). Within ALOs, individual leadership development interventions are embedded within the wider context and culture of organization-wide talent optimization and leadership integration.

The article also includes eleven principles of leadership development, which have been interwoven throughout the design of this framework.

### 1.8. Contextual Considerations

That said, while the Optimizing System is intended to guide leader and leadership development generally, it is comprehensive, and some strategies may be considered superfluous or inapplicable to certain circumstances, or will manifest differently when applied to different interventions. For example, although there is value in conducting a pre-program needs analysis and identifying a non-intervention control group, these strategies may be expedited, approached informally, or considered non-essential in advance of an informal mentoring program for a small team. In addition to its purpose and scope, distinct program contexts can influence how the Optimizing System might be applied. For example, during various stages of a crisis: escalation, emergency, recovery, and resolution, leadership development is approached very differently [99]. Other variables include single versus multiple intervention programs, formal versus informal components, whether programs are internal, open-enrollment, or community-based (i.e., invited representatives from different organizations within the same community), and whether the program is standalone versus connected to a wider talent optimization or leadership integration strategy or ecosystem.

Finally, given the contextual and adaptive nature of leadership [100], which normally takes place in dynamic and complex environments [101], different approaches are more effective in some circumstances than others. Consequently, program content should be presented and discussed from this contextual perspective: when may certain strategies be optimal and when not, and how might leaders adapt them according to the situation?

## 2. Methods

### 2.1. Research Question

The central research question guiding this study was the following: which elements of program design, delivery, and evaluation, along with application of learning strategies, can most reliably contribute to maximizing the outcomes and return on investment of leadership development?

### 2.2. Data Sources

To answer this question, we drew on the theoretical and empirical foundation established in Geerts (2024) [88], as well as four other data sources. Three were independent systematic literature reviews of leadership development for professionals whose formal research protocols [102] were guided by specialist librarians from the University of Cambridge [27,53] and Oxford University [25]. The reviews were informed by The Preferred Reporting Items for Systematic Reviews and Meta-Analyses (PRISMA) [103] and the Cochrane Review Handbook for Systematic Reviews of Interventions [104].

### 2.3. Inclusion Criteria

Primary research studies identified in English-language, peer-reviewed academic journals were included [53] if they adhered to the following:focused on leader or leadership education, development, or training programs, including individual developmental activities, such as coaching;featured adult professional participant samples, not children, undergraduates, or pre-qualified trainees, such as military cadets or medical students;reported results of an evaluation of the effectiveness of the program, not simply presenting a model, theory, or program that was not evaluated.

Programs featuring a single task, such as creating a business plan, or capability, such as innovation, or interventions where leadership was only one of many learning outcomes were not included [27].

### 2.4. Review Details

The first systematic review was conducted for an unpublished doctoral thesis [53] and included 56 empirical studies from various sectors, including the business and private sector, the military, the public sector and government, manufacturing, engineering, healthcare, public health, and higher education. The second, by Geerts et al. (2020), focused on healthcare professionals and identified 25 empirical studies, as well as featuring a novel approach [27]. This review was the first to differentiate its included studies according to the quality of their evidence into three tiered categories (bronze, silver, and gold) based on appraisal results using the Medical Education Research Study Quality Instrument (MERSQI), a validated instrument. It was also the first to isolate program elements exclusively from top-quality studies (the silver and gold categories). This critical process resulted in a collection of “gold-standard” elements of design, delivery, and evaluation (Table 1), reinforced by the best available evidence linking them to outcomes [27,105]. 

The third source was the systematic review by Lyons et al. (2020) [25], which identified 117 included studies of medical leadership development, replicated the aforementioned gold-standard methodology and categories, and added appraisals using the Joanna Briggs Institute (JBI) Critical Appraisal Tool for qualitative studies [106,107].

The final data source was a literature review specifically on training transfer guided by a robust peer-reviewed search strategy [108], which identified several meta-analyses [10,12,19,22,35,44,56].

### 2.5. Review Results

Combined, these reviews identified 172 unique studies from 2000 to 2020, including 30 high-quality studies. The collection of gold-standard elements drawn from this exclusive subset can inform further research and programming, including the choices proposed below [27].

### 2.6. Coding

To prepare for analysis, all included articles were coded meticulously. Fields included: program location, length, format (e.g., virtual), goals, provider (internal vs. external), faculty, participants and their selection criteria, curriculum, leadership models or frameworks included, developmental activities, resources, delivery details, desired outcomes, application strategies, evaluation details, and reported outcomes, as well as corresponding reported lessons. Each article was also examined for reported barriers to application and explanations for substandard outcomes [109], which informed proposed strategies to counteract these shortcomings.

### 2.7. Analysis and Framework Creation

Through an iterative, inductive process centered on the gold-standard studies, the data were analyzed for credible theory and evidence supporting effective approaches to program design, delivery, and evaluation, as well as for training transfer. The results were combined with the evidence and theory identified in Geerts (2024) [88] to form the empirical and theoretical bases for creating the framework. The Optimizing Framework was created, reviewed, and revised through multiple rounds of an iterative, inductive process, again centered on the gold-standard studies, as well as the principles of leadership development [88] (Table 2).

## 3. Results

### 3.1. A Framework for Maximizing Leadership Development: The Optimizing System

#### Structure

The Optimizing System (Table 3) includes elements of gold-standard programs and strategies for maximizing their impact and ROI. Following a foundational set (F), they are organized sequentially according to when they can be implemented: before (P = Pre), during (D), at the conclusion of (C), and sometime after (A) leadership development programs. The strategies are mainly subdivided into major categories: program design, delivery, personnel (participants and faculty), direct application strategies, evaluation, and organizational culture and support. Each of the points below is worded in a directive manner in order to avoid repeating “Research suggests that the outcomes and ROI of leadership development are maximized when providers and/or organizations, …” Preceding the strategies below, “(GS)” indicates support by gold-standard evidence [25,27,105] (Table 1) and “ideally” indicates strategies pertaining to organizations with extensive leadership integration, as in A Leadership Organization. For each strategy, the supporting evidence, role in facilitating desired outcomes, and relevant considerations are presented, along with some application examples.

### 3.2. Foundations for Leadership Development Programs (“F”, n = 9)

Foundational strategies pertain to what the organization has in place that supports leadership development programs directly or indirectly.

### 3.3. Foundational Models


**F1 (GS). Have a shared leadership model and capability framework as the common conceptualization and language of leadership**


Adopting a leadership model and a leadership capability framework is essential for defining what is meant by leadership conceptually and for describing the specifics of effective practice [88,110]. These resources can guide individual leaders’ decisions and actions and serve as a common language to enhance collaboration. Organizationally, they can help structure strategic decisions and innovation projects [111,112]. They can also inform talent optimization functions and resources, such as performance and development targets and reviews, as well as curricula for development programs and 360-assessments, and facilitate integrating leadership organization-wide, which is vital for embedding leadership in the organizational culture and becoming A Leadership Organization [111,112]. It is beneficial when they are customized to the organization and when the capability framework describes behaviors at different levels of leadership (e.g., frontline, middle, senior, and executive) [86,113,114,115]. An example of the latter is the LEADS in a Caring Environment Framework (LEADS) [111,116], which is the most widely used resource of its kind in healthcare in Canada [117,118]. Integrating LEADS organization-wide has been empirically correlated in multiple sites with improved outcomes at the individual and organizational levels [119].

### 3.4. Organizational Culture


**F2 (GS). Earn executive support for leadership development as a key strategic enabler and investment in their people**


Leadership development can have strategic value in terms of advancing strategic priorities, optimizing performance at all levels, and increasing staff engagement, job satisfaction, and retention [25,27]. Executives who are convinced of this, versus viewing leadership development as expendable or a luxury, prioritize development, allocate resources, and actively endorse participants and programs, including as sponsors and guest speakers [7,34,49,120,121,122,123,124]. The lack of senior leadership support is commonly viewed as the principal threat to the application of learning [34,49]. Edmonstone (2009) reports that the leadership program he studied was successful when there was an executive-level champion during and after the program; conversely, when the program lacked a champion and that person’s support, outcomes suffered [115]. Executives are also instrumental in leading by example by participating in development themselves [7,44,49,125], which can then cascade throughout the organization, and by shaping an organizational culture that encourages and expects leadership, experimentation, and innovation [7,49,126,127,128,129,130].


**F3 (GS). Embed leadership development as a key component of talent optimization, aligned with the overall organizational purpose, values, vision, mission, and strategy (True North)**


Leadership development is most effective when understood as a key component of talent optimization, alongside other complementary functions [88]. A robust talent optimization strategy requires a comprehensive framework [88], including talent acquisition (recruitment), talent integration (onboarding), talent specialization (identification), talent motivation (engagement, rewards, retention), performance validation (management), talent connection (social), and talent continuation (succession) [88]. These should be explicitly directed toward the organization’s True North and reviewed regularly to ensure that they are intentionally addressing current and evolving strategic, HR, and business priorities [7,44,49,70,81,114,121,122,130,131,132,133]. Understanding the role of development programs in the broader context can increase participant motivation and commitment, as well as supervisor support.


**F4. (GS) Provide funding and/or resources and protected time for leadership development**


When development interventions underperform or fail, participants frequently cite a lack of time to participate fully, reflect on and discuss their learning, and/or apply these learnings or advance change initiatives at work [7,49,53,85,114,127,132,134,135]. For example, although impact projects can be a highly effective developmental activity when properly supported [40,136,137], if participants are expected to design and implement them on top of their existing workload without sufficient protected time or clear alignment with their current priorities, then frustration, incompletion, and poor results are likely [36,44,138,139]. Securing time for leadership development, covering course costs, providing stipends, and curating a library of in-house or online resources are further examples of organizational support, which is expected when leadership is integrated organization-wide [123].


**F5. (GS) Ensure there is a safe culture of learning and leadership within the program and in the organization**


To maximize their experience of, and contributions to, programs, participants, faculty, and guest speakers must trust that the developmental environment is safe, psychologically, emotionally, physically, and culturally, and conducive to experimentation and learning [140,141]. The “environment” refers to program spaces (physical or virtual), personnel, communication, and materials, as well as to participants’ work contexts. Many important topics and issues, planned and emerging, are complex and even divisive or triggering, and thus require a tactful approach. A safe environment instills trust that one can express oneself candidly and confidentially, and that there is shared respect for all people and a commitment to supporting their development [140]. Safe learning and leadership cultures involve regular experimentation and honest and constructive feedback, expecting that mistakes and setbacks will occur and can catalyze deeper learning [92,142]. Confirming these norms at the outset can be reinforced by having participants sign written codes of conduct. Faculty are responsible for verifying that these are being upheld throughout and resolve any concerns quickly.

### 3.5. Design


**F6. Ideally, have developed a comprehensive ecosystem of leadership development interventions, experiences, and resources available to staff at all levels, aligned to career pathways**


Leadership development is optimized when instead of isolated interventions, there is a coordinated ecosystem of formal and informal development activities and resources [19,88,143,144]. These components are best organized using a framework according to their main characteristics and optimal utility. As that described in Geerts (2024), categories include individual, educational, relational, experiential, organizational, and resources [88]. Ideally, numerous options would be available to staff at all levels, aligned to their career pathways and development plans, with specific programs designated for key roles (e.g., new directors). Without such coordination, leadership development becomes a disjointed “hodge-podge of classes and lectures lacking coherence, logical progression, comprehensiveness, and relevance” ([145], p. 542). Limited programming also tends to lack equity, typically catering to a relatively small group of senior leaders, with few or no options for others. An explicit ecosystem can increase participants’ motivation and commitment by clarifying the relevance of each component to their career journeys, enhancing outcomes [12,50,69,130,146].

### 3.6. Leadership Integration


**F7. Ideally, have distributed leadership organization-wide, fully integrated into talent optimization, embedded in the organizational culture, and is an accountability expectation of all staff to develop and support the development of others**


Similarly, when leadership is embedded in talent optimization and in the organizational culture as an expectation of all staff, as in an ALO, program participants, their colleagues, supervisors, and senior leaders are held accountable for their own development and that of others [88]. For example, when leadership and development goals are included in job descriptions, performance reviews, and promotion criteria, it signals that they are prioritized by the organization and that results are expected [12,50,69,130,146]. This enhances the perceived relevance of programs, participants’ motivation and commitment to achieving their goals, and supervisor and colleague support offered to achieve success [12,50,69,130,146]. In leadership cultures, supervisors are responsible for their reports’ development, increasing the support they offer participants and accountability for both parties. Similarly, senior leaders are actively engaged in development as sponsors, speakers, or mentors. Finally, participants are surrounded by colleagues who are also regularly engaged in development and expect application and experimentation, which enhances their openness, encouragement, and support as well.


**F8. Ideally, synthesize graphically the various forms of leadership development and integration, as well as their interconnectivity to each other and to career pathways for all people, in a blueprint**


It is helpful for organizations to have a graphic summary or blueprint of all aspects of the leadership development ecosystem and talent optimization functions, including the program and personnel details, goals, and interconnectivity of each to each other and in the larger scheme. Such a blueprint reinforces that leadership development is fundamental to achieving system-wide priorities and to supporting individual career development, and is an important step in building and sustaining a culture of leadership and innovation across the organization. It is beneficial to have a subset of supporting graphics specific to various roles and career tracks (e.g., physician leaders), which can be used in career planning and development discussions. In addition to serving as a visual inventory, the blueprint can also highlight gaps in programming, personnel, equity, and support for organizational priorities.

### 3.7. Support


**F9. Develop a communications strategy for the program to relay progress and celebrate achievements**


A formal communications strategy for leadership development programs can increase the cachet and perceived importance of programs, as well as supporting recruitment, engagement, and retention strategies. It can also be used to celebrate faculty, participants, and their achievements, as well as demonstrating the organizational commitment to its people and to leadership development. Finally, these measures can heighten social accountability among personnel to achieve program and individual outcomes.

### 3.8. Pre-Leadership Development Strategies (“P”, n = 23)

#### 3.8.1. Context and Engagement


**P1 (GS). Conduct a needs, gaps, opportunities, and strategic priorities analysis involving key affected parties (stakeholders) to inform program design and to generate engagement and support**


Informing development program design by a needs, gaps, and opportunities analysis can increase their quality, perceived relevance, and effectiveness [19,51,113,123,147]. Along with a review of relevant research and leading practices and an environmental scan of emerging trends in local and broader contexts, customizing programs based on input from diverse key affected parties (stakeholders) and experts can ensure that the purpose, priorities, desired outcomes, personnel, and content of leadership development will best serve the organization as it evolves [7,19,40,52,113,130,147,148,149]. This engagement also increases the likelihood that contributors will share ownership of the success of programs [7,49,63,85,113,122,123,130,147,150,151]. Despite this evidence, the practice is rare [28]. For example, McGurk (2010) attributes a lack of substantial or long-term program impact to the design’s lack of connection to specific business needs and to the organizational context [152], reinforcing that leadership development is not one-size-fits-all [19,88,143,144].

Information can be gathered from affected parties regarding priority program focuses, participants, or strategic initiatives, as well as by examining recent organizational staff engagement survey results, 360-assessment aggregate reports, and the blueprint.

This process can indicate personnel, expertise, experiences, standards, and partnerships required to address current gaps, develop key talent, prepare people for specific roles (e.g., recently promoted leaders), increase capacity [122,148], and achieve strategic priorities, or to maximize opportunities to improve, optimize, innovate, or expand [7,28,130]. Gaps can include those in equity and diversity (EDI) representation in leadership positions and in development program faculty or participant samples.

Engagement can involve senior leaders and those in key roles [130], such as the strategy lead, HR and OD professionals, those directly impacted, such as clients or community members [78,80], and internal exceptional performers. The latter understand the capabilities and resources required to perform outstandingly [153]. Involving potential program participants in this process can also increase their motivation to develop and their accountability [7,48,154]. Similarly, engaging participants’ supervisors, sponsors [7,12,44,72,81,122,155,156,157], and colleagues [158,159,160] in advance can increase the encouragement, resources, and support they offer, as well as their receptivity to experimentation as part of a learning and leadership organizational culture [7,49,126,127,128,129,130], which are vital to maximizing outcomes [7,34,49,52,56,69,124,147]. Involving a diverse pool can also uncover any misalignment that could obstruct outcomes if not addressed.

#### 3.8.2. Design


**P2. (GS) Determine the scale, scope, and provider of the program with a preliminary consideration of return on investment (ROI)**


Determining the optimal and feasible scale and scope of programs, given the budget and internal capacity to design, deliver, support, and participate, is important. This involves estimating the cost (direct and opportunity) and ROI. This analysis informs choices of the number of participants, types, number, and format of activities, formal versus informal options [7], and whether to pilot-test prior to widespread implementation [82]. Pilot-testing can enable providers to assess the effectiveness of an intervention, to refine aspects based on feedback from faculty and participants, and to establish initial data to garner support for further development.

This also includes the decision of providers: internal, external, or mixed. In-house interventions are typically less costly [50,85], partially due to internal facilities and faculty, and are customized to the organization, referencing actual aspects of the participants’ workplace and culture, resulting in increased perceived relevance [7,19]. In their systematic review, Lacerenza et al. [19] (2017) found that in-house interventions facilitated superior outcomes and ROI compared to external programs [28,50]. However, in-house programs can be insular, lacking different perspectives and insights about, and from others within, the broader, socio-political context [57]. Likewise, some organizations lack the expertise, resources, or capacity to design and deliver programs. Conversely, external programs can expose participants to diverse faculty, perspectives, and ideas, as well as professional facilitators and modern facilities [19]. However, they tend to be more expensive [85] and may disproportionately rely on standardized, “off-the-shelf” programs [24], which can detract from their perceived relevance and utility. Mixed programs can combine benefits of both; however, they require distinct coordination to maintain cohesion and continuity among component parts.


**P3. (GS) Apply a robust outcomes-based design strategy, beginning with creating explicit program goals**


Optimal programs feature a robust outcomes-based design, such as that described by Geerts et al. (2020) [27] (Table 4) and applied to the design of the Inspire Nursing Leadership Program (INLP) by the Canadian College of Health Leaders (CCHL) and the Canadian Nurses Association (CNA) [105].

Informed by the steps above [7,27,78,80,81], design begins with creating specific program goals that are customized for its purpose and participant group [34,85,128,152,154,161] and aligned to the organization’s True North [7,49,81,114,131,132,133]. Without clear goals, there can be confusion, conflicting expectations, and poor outcomes [44,115,130,162]. Goals should be communicated to all involved prior to registration and at the outset, revisited throughout, and included explicitly in the evaluation framework [7,27,78,80,81]. It is beneficial if goals include increased participant self-awareness and self-efficacy [88], since these capabilities are proven to augment leadership effectiveness [34,163,164,165]. Self-efficacy refers to one’s confidence in having the knowledge, capabilities, experience, resources, and capacity to meet role and performance expectations [166]. Program goals should extend beyond individuals’ development to include those at broader scopes, such as the team and organizational levels.


**P4. (GS) Select ensuing desired outcomes, including level of mastery, at various levels**


Next, providers identify specific desired outcomes to achieve the program goals [7,19,35,53,94,96,130,167,168]. To categorize these outcomes, drawing on Kirkpatrick and Kirkpatrick [96], the framework described in Geerts (2024) includes levels of increasing scope [88] (Table 5). Each of these is divided into subjective or objective (verified by external sources, tangible results, or statistics) [88]. There are also three core outcomes, which can relate to various scopes: environmental (Env.); equity, diversity, inclusion, and accessibility (EDIA); and economic (Ec.) impact (direct or indirect), such as cost savings [88]. 

Within individual-level outcomes, consideration should be given to the degree of mastery, which, drawing on Dale (1969) [94], is divided in Geerts (2024) into mastery of concepts (knowledge) and mastery of capabilities. 

Concept mastery progresses from the ability to remember, understand, analyze critically, apply in theory or practice, reflect critically, contextualize, teach with authority, create new content, validate and refine, and reproduce the creation and validation process successfully [88]. 

Capability mastery progresses from the ability to remember, apply practically (and receive feedback), reflect critically on experience and feedback, be able to adapt approaches, practice and develop as a habit, contextualize and understand why different approaches are effective or not, analyze critically, teach with authority, create new content or approaches, experiment, validate and refine, and reproduce the creation and validation process successfully [88].

Naturally, different levels of mastery require different design, delivery, and evaluation approaches.


**P5. Ensure that diversity, equity, inclusion, and accessibility (EDIA) are prioritized in the selection of participants, faculty, speakers, and content**


Diversity is essential for ensuring programs are appropriately equitable and inclusive and for maximizing their effectiveness. Diversity includes gender, race, age, sexual orientation, and people living with disabilities or neurodiversity, as well as profession, stage of career, geographical location, sector, etc. Thus, it combines the diversity of identities, backgrounds, experiences, and perspectives. This priority builds on evidence that effective diverse teams outperform homogenous ones [169,170,171,172,173,174,175,176,177]. Providers should ensure that diversity is visibly reflected in program participants, faculty and speakers, and course materials (e.g., visuals and examples), and is particularly reflective of the staff and beneficiaries/community populations.

#### 3.8.3. Participants


**P6. (GS) Select participants intentionally to address organizational needs and priorities**


Selecting program participants should be informed by the needs, gaps, and opportunities analysis and directed toward organizational and talent optimization priorities [128,150,178,179]. The latter can include onboarding new leaders, supporting executives to lead the system, or preparing high-potentials for promotion as part of succession planning [50,122,130]. High-potential employees are those who have the experience, expertise, and leadership to likely succeed in advanced roles [180,181]. Choosing people wisely is important, since the Edmonstone (2009) study reports that inadequate participant selection is a key contributor to program underperformance [115].

Another selection consideration is having teams or groups from the same organization or community participate in leadership development together [7,19,32,49,82,182,183,184], building on the distinction between (individual) leader development and (collective) leadership development described above [13]. The meta-analysis by Salas et al. (2008) found 20 per cent higher team performance outcomes resulting from team versus individual training [185]. Similarly, Dannels et al. (2009) reported that the impact of leadership development was significantly higher in organizations from which three or more participants attended than in those from which fewer took part [186]. These findings are perhaps attributable to participants developing their teamwork and leadership capabilities together using common language, feeling enhanced relevance to their shared workplace, and holding each other accountable.


**P7. (GS) Address participants’ motivation to learn and ensure that they can commit fully to the program with supervisor and sponsor support**


Participant investment and capacity to participate in leadership development interventions are important pre-conditions of the principles of adult [85,93,187] and professional learning [88] and major predictors of outcomes [7,12,13,24,93,159,188,189,190,191,192,193,194,195]. Participant motivation is influenced by their perception of the program’s utility and relevance to their passions, accountabilities, job performance, and career pathways, which are enhanced through personalization of goals and projects [7,44,196]. Participants’ commitment and engagement are equally important, since without full attendance, assignment completion, and in-session participation, individual and collective learning is diminished [48,145]. Upholding attendance and participation requirements can counteract this; however, it is most effective when supervisors temporarily modify participants’ workloads to allow time to apply their learning [36,44,138,139]. Motivation and commitment are increased when participants must apply to programs and be selected, as well as when supervisors and sponsors nominate or personally endorse them, check in regularly regarding progress, and attend important sessions (e.g., first and last) [85,187].


**P8. (GS) Have participants create a Leadership Development Plan (LDP) that includes personalized goals and desired outcomes, aligned to their organizational plans and career pathways**


Having a program Leadership Development Plan (LDP) enables participants to record and commit to personalized goals and desired outcomes aligned to their career aspirations, as well as helping them organize their learning and track their progress [86]. The personalization of program goals, projects, and desired outcomes can increase participants’ perceived relevance, motivation and commitment, and impact [7,44,82,93,113,128,152,197,198]. For projects, when participants consult with non-program collaborators and beneficiaries regarding the need, approach, and metrics beforehand [105], outcomes are enhanced [36,85,188,199]. Goals can be optimized when they are SMART: specific, measurable, attainable, results-based, and time-bound [200]. These goals are optimized when aligned accountably to participants’ workplace development plans, performance metrics, and career pathways [12,50,69,130,146].

Though rarely implemented [53,152], personalizing program goals is also important because of the proven impact of several individual characteristics on application outcomes. These include participants’ innate orientation to learning and goal achievement [195], their baseline proficiency of leadership capabilities and their capacity limits [201], their ability to develop when challenged [190], and their perceived control over their own development and achievement of outcomes [188]. Additionally, having participants’ supervisors approve their goals can verify their feasibility and result in additional support to facilitate their successful achievement [7,198].


**P9 (GS). Ideally, align programs to internal and/or external professional requirements (e.g., Maintenance of Certification), distinctions, or credentials**


Program credibility and participant motivation and commitment are increased when successful completion “counts toward” internal and/or external professional requirements, distinctions, and credentials, linked to their career pathways [105,202]. The former can refer to routine organizational mandates or those required for advancement, or externally to satisfy annual Maintenance of Certification for professional bodies. Similarly, interest is often heightened in programs linked to earning distinctions, such as diplomas, certificates, university degrees, or professional certifications, such as the Certified Health Executive (CHE) designation offered by the Canadian College of Health Leaders [105,202].

Organizations must balance the appeal of individual distinctions with equity, resource, and capacity considerations. That is, highly credible programs are typically expensive, highly time-consuming, and consequently, only available to a select few. Thus, it is important to ensure that organizations provide options more broadly that are feasible given limited funding, internal capacity to design and deliver/implement programs and tools, and staff capacity to participate, given their workloads and required coverage. One solution is having a wide range of formal and informal leadership development options, including structured on-the-job learning, available to everyone and built into talent optimization requirements, and to design a mechanism or passport to track how each activity counts toward credentials.

#### 3.8.4. Design


**P10 (GS). Select evidence-based program details (content, faculty, developmental activities, resources, structure, etc.) intentionally to achieve identified targets, customized for the participant group**


To maximize effectiveness and efficiency, program details should draw from the best available theory and evidence and be customized according to the needs analysis, identified targets, and participant group [7,27,78,80,81,105]. This approach has been proven to increase program outcomes considerably [7,203]. Organizations lacking the capacity to review the evidence comprehensively can reliably select program details from the gold-standard elements identified by Geerts et al. (2020; 2024) [27,105] (Table 1).

Customization can cater to participants’ geographical location, domain or sector, profession, organization, level of leadership, role, and previous leadership development experience, as well as the size of the group [86,110,113,115]. For example, one would expect that a program for international chief executive officers (CEOs) would differ from one for newly hired frontline supervisors in a small organization. Gathering input beforehand from top performers in the same area or facilitators with relevant experience can provide insight into content, program formats, and delivery techniques best suited for the participants. The design selection process should involve outlining explicitly how each aspect of the design, delivery, and evaluation contributes to achieving program goals and desired outcomes [7,27,35,85,93,94,127,130,147,168].

#### 3.8.5. Faculty and Guest Speakers


**P11. (GS) For faculty and guest presenters, prioritize diverse, mixed teams (internal/external and experts/practitioners)**


The best programs feature credible and skilled faculty, mentors, and guest presenters with diverse expertise and backgrounds. Lyons et al. (2020) identified a statistically significant correlation between programs with mixed faculty in terms of (a) internal and external to the organization, and (b) experts and practitioners in achieving organizational outcomes [25]. For both, the faculty should have established credibility that resonates with the participants.

Having internal faculty demonstrates their commitment to development [7,44,113,123,148,151] and can enhance the perceived relevance of interventions to the shared professional context [124]. Their first-hand experience enables them to address underlying organizational norms and culture, including their protocols, challenges, and resources, as they affect leadership, in a way that external facilitators cannot [11,113,128]. Additionally, participants report appreciating the value of networking directly to internal senior leaders, a benefit which extends after the program [110,113,204]. Involving internal outstanding performers with relevant roles and expertise, including program participants and graduates, can augment the quality and relevance of the content, as well as the prestige of the programs, and aligns with the principles of adult [93] and professional learning [88].

Conversely, external faculty can offer professional facilitation, prominent experience, and diverse perspectives, which can enhance the prestige of programs, broaden and deepen their content, and avoid insular thinking, though they are typically more costly [11,125,205]. Mixed faculty can draw on the strengths of both approaches, and a diverse variety offers assorted perspectives, the value and utility of which may resonate differently with individual participants.

#### 3.8.6. Curriculum


**P12. (GS) Customize the curriculum according to pre-program engagement and the participant group, and embed the leadership model and capability framework**


Customizing program content based on the needs analysis, the participant group, and their professional context increases the perceived relevance, utility, and outcomes of programs [34,85,128,152,154,161]. The curriculum involves topics (knowledge), skills, behaviors, capabilities, and perspectives that participants need, along with pressing issues in the internal and external professional context [7,35,44]. Although some leadership topics and capabilities appear to be universal, such as communication, the Lyons et al. (2020) [25] systematic review found no correlation between specific curriculum content and improved program outcomes [7,48]. This reinforces the importance of customization versus off-the-shelf offerings [19,143,144], which is amplified by considerations of different desired degrees of mastery [88]. Similarly, content should be prepared within the dynamic and complex leadership perspective [100,101]. Research suggests that management competencies, or “hard skills”, such as business acumen and delegating, appear to be easier to learn than leadership capabilities, or “soft skills”, such as adaptability or emotional intelligence [206].

Customizing the content for the participant group should involve selecting materials, examples, and resources from contexts to which participants can relate, whether similar or intentionally contrasting to avoid insular thinking [93]. For the latter, facilitators should allow time for reflection and discussion about how external examples could apply locally. The more senior the participant group, the more the focus should shift from technical management and one-on-one interpersonal skills to more strategic systems of thinking capabilities, broader innovation, and the development of coalitions with those outside one’s organization and sector. For organizations that have an internal leadership model and capability framework, they should be interwoven throughout the program to demonstrate continuity and applicability. Finally, program design should consider what other development opportunities participants are undertaking and relevant talent optimization functions that could be referenced for enhanced cohesion and relevance. For example, participants could be invited to discuss issues raised in class with their (non-program) mentors or consider how they relate to their organizational development plans.


**P13. Reserve in-program time for emerging issues and topics**


Even with robust curricula, important topics or issues may arise during interventions that appear valuable to address in class. These include topics participants feel are important, which are either not covered in the curriculum, or are worthy of deeper exploration. They can also involve pressing contextual issues, such as new government legislation or organizational restructuring, which can serve as real-life, real-time case studies to scrutinize. Incorporating these emerging priorities can augment participant learning and maintain its relevance and applicability. Time can be reserved formally in synchronous sessions, through support components, such as coaching, or in discussion forums. For example, the final module of a year-long executive leadership program by the Telfer School of Management, University of Ottawa, includes three concurrent master classes, the topics of which are only selected midway, based on participant preference.

#### 3.8.7. Design: Developmental Activities


**P14. (GS) Select developmental activities according to their intended impact on desired outcomes and offer a variety**


Programs are most effective when their developmental activities are evidence-based and selected specifically based on their utility, individually and collectively, to achieve program goals, given the participant group [39,53,94,128,132,147,148,152,159,204,207]. Activities vary in their efficacy to facilitate different types of outcomes and the extent to which they link directly to workplace application. Some activities serve to enhance others or work best in combination, such as executive coaching debriefs following multi-source feedback (MSF) [19,208]. Finally, offering a variety of activities, especially with some optionality, is beneficial in accommodating different learning preferences [7,19,94,113,114,115,122,128,152,161,207,209].

The framework by Geerts (2024) provides five categories of development activities according to their central features, purposes, functions, and connection to application [88,105]. The categories are as follows: *Individual* components that provide focus and enhance self-awareness, structuring, such as goal setting and career planning [210], diagnostics, such as psychometrics, 360-assessments [38,210], performance feedback, reflection [182], and credentialling [202]. *Educational* components, typically involving presented didactic content, include lectures [136,161], guest speakers [147], site visits, and case study analyses [147]. *Experiential* activities, involving practice in a safe environment [60], such as role play and simulations [136], workplace application exercises [13,36,41,53,85,136,147,198,211,212], and leadership impact projects [40,136,137]. *Relational* learning, which results from dialogue [110,122,128,147,162,204,207,213,214] and includes in-class or forum discussions, mentoring [128,215], coaching—individual [38,40,137] and peer [216]—networking [217], and communities of practice [218]. Coaching and mentoring are particularly important during longer programs by increasing participant focus, accountability, and self-awareness, by providing extra learning insights, and by navigating challenges experienced in application [11,86,114,128,148,207,213,219]. Bowles et al. (2007) demonstrated that participants who received coaching achieved significantly higher results than their non-coached counterparts [213]. In-class activities can be reinforced by *resources*, such as articles and multi-media clips, with clear expectations of pre-work and asynchronous tasks, such as required readings and posting in online forums.

Although formal leadership programs often default to traditional lecture formats [39,183,184,220], multifaceted interventions with experiential components, coupled with structured on-the-job learning, are becoming more common [27,34,113,195,198,221].

Finally, designers should consider whether artificial intelligence and other forms of technology can be incorporated beneficially, the advantages, limitations, and feasibility of which continue to evolve [7].

Therefore, selecting the optimal package of experiences, resources, and tools for the purpose, goals, formats, and participant group is essential for maximizing program outcomes.


**P15. (GS) Embed activities in reflective or experiential learning cycles (RLCs and ELCs)**


The impact of some developmental activities can be maximized when they are embedded in reflective or experiential learning cycles (RLCs and ELCs), drawing on Kolb (1984) [97], which require designated time for each component [7,35,49,53,88,97,113,123,222]. Both these cycles involve sequential processes with a developmental activity in the center, framed before and after by strategies to optimize learners’ development. Reflective learning cycles (RLCs) can apply to all activities, whereas ELCs pertain to experiential activities and involve performance feedback.

The sequence is as follows: initial interest/questions, goal setting, the activity or experience, performance evaluation and feedback (ELCs only), discussion, reflection, revision of goals, support, and repeat (ELCs only). This process accelerates the rate at which participants learn from experiences and progress toward mastery [7,19,82].

To sharpen focus and increase participants’ motivation and commitment, the cycles start with exploring their interest in the activity and potential questions, which prime them to engage fully. Next, they set activity learning goals, aligned to their program goals [7,223], and then experience the activity. For ELCs, after and, when appropriate, during experiential activities, such as simulations, participants use pre-defined criteria to self-assess their performance, which is a key leadership skill [7,44,224,225,226,227,228], and receive feedback from fellow participants, program facilitators, and/or workplace colleagues [7,19,35,65,82,130,229,230]. Feedback should be timely, considerate, constructive, and actionable [7,231,232]. Comparing one’s self-ratings to others’ can gauge their congruence, enhancing self-awareness [225,233,234]. Analyzing and discussing video recordings of these experiences can be highly beneficial [161,235,236,237]. Although excessive evaluation should be avoided [44], the effectiveness of experiences with no feedback appears to be comparatively reduced [113,130]. For all cycles, debriefing the feedback received and/or the experience of the activity with peers and/or facilitators enables participants to identify outstanding questions, capabilities to further develop, strategies that were effective (or not), and potential next approaches [7,229,238].

Participants should also be granted time to reflect on their experience, feedback, and lessons, as well as their relevance to their work context [75,226,239]. This exercise serves as an additional heuristic developmental tool [198,240]. For example, the most consistent request by participants following a week-long executive leadership development program, run jointly by The Staff College: Leadership in Healthcare and the UK Defence Academy, was for more time to reflect on the course content [241]. This stage is also an opportunity for participants, in consultation with faculty and/or sponsors, to re-evaluate and consider adapting their program goals [65,93,95].

Finally, for ELCs, this cycle can be repeated, since practicing skills and behaviors, even after correct performance, deepens capacity based on varied experiences and creates automatic responses, the latter of which frees cognitive energy, which can be dedicated to more complex tasks [7,242]. This repetition can be enhanced by gradually increasing the difficulty of the conditions and decreasing facilitator support, leading to greater self-efficacy [7]. This process relies on a psychologically safe space in which participants can experiment, commit errors, and receive follow-up support that enables them to develop their knowledge, skills, adaptability, and resilience [140,141].

Reflective and experiential learning cycles can be embedded in programs in a myriad of forms, with some steps being prioritized and others included informally or omitted based on their anticipated utility in maximizing activity impact.

#### 3.8.8. Direct Transfer Strategies


**P16. (GS) Ensure that application/transfer is included explicitly throughout**


Embedding the application of learning to work in intervention designs is essential for consolidating learning, progressing toward mastery, and maximizing program impact [19,49,53,97,237,243]. When not built in, research has shown consistently that transfer is limited and that participants typically revert to previous behaviors [44,132,138,244,245,246]. The experience of applying learning on-the-job throughout development programs also validates and deepens participants’ understanding and efficacy, given the complex nature of leadership. Validation is essential, since the military expression “no strategy survives first contact with the enemy” indicates that no training environment can replace lessons learned in real conditions. For example, when participants implement projects during the program, the process reveals insights beyond taught content, which can be discussed in class to contribute to overall program learning. Post-application discussions also enable fellow participants and faculty to provide support or propose alternative approaches when challenges are encountered, an opportunity not afforded when application is left until after the program [7,81,114,147]. In addition to lessons participants can learn and capabilities they can develop from workplace application, the process can also enable them to enhance their relationships with colleagues and understand their priorities and readiness for change. Finally, application, particularly through projects, can also provide evidence of outcomes and return on investment, compared to a project proposal intended to be actioned following the intervention.

#### 3.8.9. Design


**P17. (GS) Select the optimal program structure and format**


Optimal designs involve logistics (structure and format) determined specifically to achieve program goals, given the details above. Aspects include the program location (in-house, external, or mixed), format (in-person, virtual, hybrid, or blended), scheduling (synchronous, asynchronous, or mixed), social nature (communal or self-directed), length, and optionality. For each aspect, it is presumed that “mixed” could involve combinations of the alternatives presented below.

In terms of location, in-house programs avoid facility rental costs and travel time and expenses, while also featuring the physical spaces and equipment from the participants’ workplace, increasing relevance [7,19]. Off-site interventions, on the other hand, can minimize work-related interruptions, enabling participants to fully engage [19]. Suutari and Viitala (2008) found no difference in outcomes when comparing in-house to external programs [195].

Format-wise, in-person interventions can increase participant engagement and experience, can be more effective for experiential, creative, and interpersonal activities, and allow for informal interactions (“watercooler discussions”) and networking opportunities. They also minimize technological interruptions. Virtual interventions are typically less costly, more convenient and less disruptive scheduling-wise, and can involve larger numbers of participants from a wider geographical range. The latter can be important from an EDI and physical accessibility perspective, although access to and familiarity with technology can be an issue. Hybrid (in-person and virtual, simultaneously) aspects can be inclusive, accommodating those unable to attend in person for planned or unexpected reasons; however, this format can involve double the effort to design, coordinate, and facilitate both groups concurrently, with reduced effectiveness and participant commitment to attending in person, given the virtual option. A preferred alternative is a blended format, involving non-overlapping in-person and virtual components with intentional choices about which components are in which format.

Further format considerations involve which components are synchronous, which feature shared experiences, discussions, and social accountability, versus asynchronous, which are more convenient but rely on pre-prepared content and parallel interactions. Similar benefits and limitations exist with cohort-based programs, with groups participating together, versus individual, self-directed approaches.

In terms of the length of leadership programs, Lacerenza et al. (2017) note a linear, positive relationship between longer programs and improved outcomes at individual and organizational levels [198]. The authors suggest that demonstrating cognitive, behavioral, and organizational changes takes time [247,248,249], as does progressing toward mastery [23,88]. Longer programs also support the implementation of impact projects, which necessarily produce outcomes [41,212]. Lengthy programs risk participant workload stress and cognitive overload [44,250], which can be reduced by spacing sessions [12,19]. Spacing, along with providing reflection time, increases the rate and volume of information retained, as well as enabling participants to apply their learning in between sessions [198,239,251]. Conversely, the effect sizes for shorter programs are reportedly smaller [12,35]; however, they can be effective for task-based, micro-burst, or just-in-time training [32,183].

Finally, designers must determine whether any program components are optional. Optionality can increase participant motivation and engagement, since they attend only sessions that interest them; however, it can increase design and administrative work to coordinate, reduce cohort continuity and commitment, and deprive participants of exposure to important content that they skip.

#### 3.8.10. Delivery


**P18. (GS) Incorporate the principles of leadership development and of adult learning**


To optimize development programs, the design and delivery should align with the principles of leadership development [88] (Table 2) and with Knowles’s (1984) principles of adult learning [86,93,95,115,154,187,252]. The latter involves addressing their pre-program motivation and commitment to learn, the strategies for which have been described above [85,187]. This is further enhanced by self-directed components and customization [93]. The first principle involves facilitators actively drawing on participants’ expertise as a source of valuable content during discussions and by providing feedback [93]. Second, faculty should link new learning to participants’ existing knowledge and propose how it can be extended and applied to their work [93]. Third, program content should focus on its relevance to participants’ professional context [93], which is verified through background engagement and analysis, customization of the design, and the selection of appropriate faculty and speakers. Finally, professional learners prefer outcomes-based programs [93], which is the basis of this framework.


**P19. For programs involving mixed participant samples, include designation-specific cohorts or syndicate sessions**


There is ongoing debate regarding whether participant samples in leadership development programs should be specific to designations, including their sector, organization, profession, level of seniority, and roles, or whether the groups should be mixed [27].

Many open-enrollment leadership programs at business schools and private providers involve mixed samples, which can attract enough numbers at premium rates to afford high-profile faculty, advanced technology, and modern facilities, as well as exposing participants to diverse peers and perspectives. Mixed, in-house programs enable participants to learn with colleagues outside their usual circles, which can provide a valuable understanding of and relationships with those in other roles or departments, as well as potentially increasing collaboration and teamwork [32,36,85,134,136,162,186,209,253].

Conversely, supporters of designation-specific samples, such as executive- or physician-only, contend that the commonalities among participant backgrounds, contexts, experiences, and scope of responsibility, along with the corresponding design customization, positively influence program outcomes [24,86,113,115,147,215,239]. This is partly attributable to a shared trust and sense of psychological safety among similarly qualified peers, which increases their openness to be vulnerable in a way that they would not in mixed groups [239]. Similarly, some programs are reported to have failed, at least in part, because of mixed samples. For example, although mid-level participants in the study by Pradarelli et al. (2016) described several benefits of a leadership intervention, many senior-level participants stated that it had no impact, suggesting that including junior-level participants (residents) diluted the caliber of the program [86].

An alternative is embedding designation-specific cohorts or small group sessions within mixed-sample programs. These sub-groups could provide an extra peer support network [7], in which those facing similar professional demands can discuss workplace challenges and can problem-solve collectively in a trusted environment [254,255].

#### 3.8.11. Evaluation


**P20. (GS) Develop a robust evaluation framework for the program, including the ROI, and for participants, and consider a control group**


Program evaluations typically only assess participant satisfaction [57], but a robust outcomes-based evaluation framework is essential for maximizing program impact [11,12,21,34,122,130,198,256] and for demonstrating ROI [257]. Evaluating the program overall and its components in reference to their goals can inform quality improvement during and following the intervention. A formal evaluation framework also reinforces that affected parties value the program and expect that outcomes will result from it [130,223,258]. Similarly, making faculty and participants aware beforehand of metrics at different levels can enhance outcomes by providing focus, motivation, and accountability [12,34,198]. Tangible participant outcomes, such as cost savings generated through impact projects, can be used to calculate the program ROI by measuring them against the expense and opportunity cost of the intervention [257].

Assessment should involve collecting various forms of data (formal, informal, subjective, objective, qualitative, and quantitative) from multiple sources at the outset, throughout, at the conclusion of, and six to nine months after interventions [27,34,80,130,145,259]. An example of such a framework in practice is described in Geerts et al. (2024) [105].

Formal evaluations are important for maintaining program credibility, quality standards, and accreditation, when applicable, and for justifying the investment to affected parties. Informal assessments can offer providers, faculty, and participants insights into successes and areas for improvement.

Qualitative data through free-text responses can provide elaborations on performance ratings/feedback and descriptions of how participants have applied their learning to work, as well as of unanticipated outcomes or those not easily quantified, such as increased confidence [105]. They can also allow for nuances of in which ways and to what extent interventions and components were effective (or not) [34,51,74,260]. Quantitative data through Likert-scale ratings and enumerable outcomes enable quick assessment, analysis, comparisons, and discussion, and can minimize interpretive bias, enhancing the reliability of the results [261].

Subjective data (perceptions) include participant self-ratings and self-reported outcomes, whereas objective data can involve external ratings, statistics, or externally verifiable results, which enhance their credibility. Combining these forms offers deeper and more complete insights than singular approaches [262,263,264].

Similarly, in terms of raters, balancing participant self-ratings with those of others, such as fellow participants, program faculty, and workplace colleagues or supervisors, provides additional opportunities for learning and self-awareness, and increases the credibility of the results, since self-reports alone have been shown to be unreliable [12,34,35,52,53,265].

Finally, although rare, it is useful to identify a non-intervention control group of individuals, teams, or organizations (clusters) that is as similar as possible to the participant group so that the key differentiating variable is the latter’s involvement in the program. Administering the same assessments to both groups and collecting identical data to track the participant group’s relative progress can strengthen claims of program causal effect and minimize speculation concerning confounder influence [263].


**P21. (GS) Establish relevant baseline measures**


It is important to establish baseline figures linked to program and individual goals to focus goal setting, measure desired outcomes and participant development, gauge program impact, and calculate ROI [27,161,248]. Clarifying specific starting points provides benchmarks for further assessments, as well as focus for participant goal setting and discussion. This process involves considering if pre-existing data can be used, such as recent participant multi-source feedback (MSF) or organizational engagement survey results, and which should be gathered specifically for the program. A similar consideration is which data should be standardized across the participant group for comparative and reporting purposes, and which should be personalized. Data can include subjective (self-reports) and objective items at the individual and organizational levels, as well as those related to application projects [121,130,131]. Individual-level measures can include participant self-ratings of knowledge, skills, or behaviors, recent performance reviews, or other internal talent optimization data. Organizational measures can include staff engagement survey results, turnover rates, etc. Finally, data relevant to projects can include performance metrics, such as recent user satisfaction scores or average service wait times. When applicable, similar data should be gathered from the control group.

#### 3.8.12. Direct Transfer Strategies


**P22. (GS) Conduct a barriers and enablers assessment and apply results to remove or circumvent obstacles and leverage enablers**


Though rarely conducted before development programs [53], a barriers assessment involving input from key affected parties and participants can highlight workplace cultural obstacles that could inhibit achieving desired outcomes [7,44,63,128,130]. Providers can use this information to circumvent impediments and increase program success. Barriers can include conflicting internal priorities, policies, or metrics; inflexible processes or practices that inhibit change; lack of senior leadership support; insufficient resources; or inadequate time for participants to implement changes [114]. Pre-program assessments should also investigate enablers in terms of organizational strategic priorities, initiatives, personnel, networks, and resources that can be leveraged to maximize results. Other enablers include talent optimization functions and non-program development opportunities in which participants are involved, which can be referenced as complementary resources.

Verifying that participants’ colleagues and workplace environments are conducive to application by circumventing barriers and leveraging enablers is vital to maximizing outcomes [7].


**P23. Communicate explicitly the purpose, goals, content, outcomes, and evaluation framework of programs overall and of individual components to establish a shared understanding and accountability among providers, faculty, participants, and other affected parties**


To maximize programs, all personnel must be aware of, and formally commit to, the program goals, desired outcomes, and evaluation metrics [34,65,114,115,122,130,162]. Similarly, it is important that faculty and guest speakers are informed of the program details, including which sessions precede and follow theirs, as well as how they will be evaluated. Providers may also choose to share program goals and highlights of outcomes and evaluation results with key affected parties, such as participant sponsors, organizational senior leaders, and a governance committee, when in place.

Communicating the goals and results transparently can activate the social contract effect and increase results [65,114,266]. A social contract can increase participant motivation and accountability to achieve goals because of implicit pressure they feel to report positive progress, even to those without formal authority over them. Engaging others in this way can also augment the support they offer in terms of advice, networking connections, or other resources.

Finally, participants can be primed from the outset for an end-of-program summary presentation of their key program learnings, outcomes achieved, and further steps, which is described below.

### 3.9. During-Leadership Development Strategies (“D”, n = 17)

#### 3.9.1. Direct Transfer Strategies


**D1. Have participants select and engage accountability teams**


Participant commitment is enhanced when they share program details and their individual goals and desired outcomes at the outset with a trusted accountability team [105]. Teams can comprise their workplace supervisors or sponsors, fellow participants, and non-program colleagues [105]. Program designs can include designated checkpoints throughout the program when participants are expected to share their learning and results with their teams and discuss how to optimize program outcomes [105]. In addition to increasing participant commitment to achieving results through this process, feedback from their teams can also enhance their learning.

#### 3.9.2. Design


**D2. Host a program launch and orientation event with senior leader support to level set and introduce personnel**


Whether as a standalone event or at the beginning of the first session, it is helpful to have an official program launch and orientation with participants, faculty, and, when possible, sponsors and organizational senior leaders. This event is an opportunity to review the program goals, details, expectations, reminding participants of their commitment, resources and how to access them, and housekeeping items, as well as enabling personnel to meet. Faculty can also introduce important supports, such as the Leadership Development Plan (LDP), journals, accountability teams, and the Community for Practice.

The presence of organizational leaders reinforces the organizational investment and demonstrates the value that they place on the program and its people. They can also provide their perspectives on the relevant organizational context: why this (leadership development and the program specifically), why you (participants), and why now (the program’s strategic importance in the evolving organizational context)? Vocal support from senior leaders, as well as their implicit endorsement of the expectations, increases participant motivation, commitment, and accountability, and improves program outcomes [7,44,49,70,81,114,121,130,131,132,133,267].


**D3. (GS) Provide networking opportunities with program colleagues and faculty, internal senior leaders, guest speakers, past program graduates, etc.**


Networking events with colleagues, faculty, guest speakers, and past program graduates are important for several reasons. First, social functions allow for one-on-one or intimate conversations among participants and with faculty and guests beyond the time allotted in class. These connections may become or deepen relationships, including with those outside one’s existing circles, which can lead to collaborations or coalitions that outlive the program [128,134,147]. Second, interactions in less-formal settings can enable participants to broaden and consolidate their learning by articulating their thoughts, hearing the perspectives of others, and together discussing feedback, issues, challenges, and outstanding questions, as well as teasing out implications and potential applications in dialogue [128,134,147]. Third, this type of environment appeals especially to those uncomfortable or unwilling to be candid in front of the full group. Fourth, providers and faculty can also gather opportune informal feedback from participants and fellow faculty regarding their program experiences and suggestions for optimization. Finally, even in a relaxed setting, describing one’s goals, progress, and strategies can activate the social contract effect and its advantages, since listeners are likely to inquire later regarding progress [65,114,266]. Interestingly, although participants cite “networking benefits” as an additional reported outcome in post-intervention evaluations, it is rarely identified as an explicit pre-program desired outcome [53].

Providers can organize official networking events during or immediately after class time, though this requires administrative work and often extra costs for venues and/or catering, which the hosts are normally expected to cover. Alternatively, participants can be encouraged to self-organize and be welcomed to advertise in class; however, attendance may suffer. A third option is treating meals included in the program, such as lunch for full-day sessions, as networking time. Regardless, social time together outside of class is valuable.


**D4. Immerse in an internal Community for Practice (when available) to discuss, share resources, and connect more widely**


Similarly, when the organization and/or the provider has an online Community for Practice, immersing personnel in it as part of the program offers a forum for asynchronous announcements, questions, and discussions; for sharing, requesting, and housing resources; and for connecting with others in the wider community. The latter can involve mentoring relationships or peer support or coaching units, or can be used as a medium to share desired outcomes and projects that may interest others. The benefits of this immersion can extend beyond the program conclusion. Creating private subgroups for each participant cohort and for all program graduates can help create community and sustain group solidarity.

#### 3.9.3. Delivery


**D5. Communicate to faculty and participants the purpose, relevance, rationale, and evaluation of each session, as well as the connection to other sessions and components**


To maximize the experience and outcomes of sessions, all personnel must be aware of their purpose, goals, details, and rationale, including their connection to other program components [34,65,114,115,122,130,162]. The purpose means the session’s role in achieving the overall program goals. Advanced relevance includes linking program content and tools to important internal resources or initiatives, such as the launch of a new performance management system. Since maximizing participants’ time and development is important, the rationale includes why design choices were made for given sessions and their goals. For example, to explore innovation, faculty might justify the time, disruption, and expense of visiting a leading-practice workplace by describing its experiential merits, versus a virtual tour or inviting a representative as a guest speaker. Providers should also meet with faculty beforehand to discuss this information, as well as the details of the participant group and their context, and any relevant issues that have arisen thus far. Faculty can then prepare their materials to align to the program and session goals and to intentionally build on, and avoid repeating, companion sessions. Providers should also confirm the faculty’s approval of the details of the components for which they are responsible before distributing them to the participants.

Materials should be consolidated in efficient packages and circulated in advance. Information can include summary agendas with topics, timings, and faculty, as well as a fuller collection of individual session descriptions and their corresponding learning objectives, faculty details, and supporting resources. At the beginning of each session, facilitators should review key details with the group in terms of its topic/focus, facilitator(s), aims, rationale, success metrics, and how it fits with previous and subsequent sessions. Clarifying these points enhances the perceived relevance of each session and eliminates confusion, questions, and related distractions [7,44,65,130].


**D6. (GS) Maximize participant engagement and hold them accountable for attendance, in-class participation, and application/asynchronous exercises**


Maximizing personnel experience and program impact requires full participant engagement in all aspects. Faculty are responsible for maintaining this standard consistently, since if engagement is low overall or limited to certain participants, outcomes will suffer. Expectations include attending all required sessions on time, avoiding distractions while in session, concentrating on the task at hand, contributing regularly to discussions and activities, and completing asynchronous exercises regularly. Even dedicated professional learners often need some benevolent management to prioritize program requirements and resist allowing pressing work or life demands to dominate or distract their attention. Similarly, without facilitation, some participants will remain quietly in their comfort zone, happily observing as more outspoken participants dominate discussions or activities. When participants fail to meet participation standards and faculty neglect to hold them accountable, typically to avoid conflict or making people feel uncomfortable, it sends an implicit message to the group that engagement is optional. This then has a spillover effect to others, which is challenging to reverse. Naturally, extenuating circumstances arise and, in some cases, being unwaveringly rigid without reasonable empathy can be inappropriate and can corrode the trust and sense of safety in the learning environment. Faculty must aim to establish a heathy balance.

Ideally, striving for intrinsic motivation and self-imposed accountability is best, rather than punitive tactics. One approach is the pre-established transparency of results, such as sharing attendance reports with participants’ sponsors or supervisors, which may prompt a discussion between them regarding what support is needed to improve it. Another is social accountability, such as scheduling to have participants discuss required readings in pairs or small groups, which might motivate them to read them to avoid the embarrassment of having to admit that they have nothing to add. Similarly, requiring all participants to prepare a question for a guest speaker beforehand, knowing that they may be selected to ask theirs in front of the group, can facilitate wide airtime distribution without surprises or relying on the regular volunteers.

Inasmuch as participants must be held accountable for their engagement, faculty must also create the conditions for all to feel comfortable participating fully. This involves clarifying expectations so no one feels blindsided, maintaining the standards consistently, and accommodating different ways of contributing. Upholding accountability demonstrates respect for the importance of leadership, the development process, the participants, and the desired outcomes.


**D7. (GS) Demonstrate a personal interest in all participants and their individual learning and development, as well as actively incorporating their expertise**


The customization of programs according to the participant group is enhanced by faculty getting to know them individually and demonstrating care and commitment to supporting their development and career goals. This also involves respecting their expertise and actively drawing on it, whether ad hoc during relevant discussions, to provide performance feedback following an activity, or officially to facilitate a session. Incorporating participant expertise as a program resource demonstrates appreciation for the caliber of the participant group and aligns with the principles of leadership development [88] and adult learning [93].


**D8. (GS) Embed discussions of experiences and lessons from application exercises regularly to enhance and extend learning, as well as to promote accountability**


Along with valuing participants’ professional experience as an important source of relevant information, faculty should incorporate participant reflections and lessons following application exercises as part of the program content. These discussions with similarly qualified peers can augment and deepen their learning [7,229,238], reinforce the contextual nature of leadership, and increase participants’ accountability.


**D9. Highlight emerging key themes, learnings, tensions, and ongoing questions throughout through different lenses**


Throughout the program, faculty should frame, link, and summarize noteworthy points in the learning journey by identifying common themes that arise, as well as overarching lessons, tensions, and outstanding questions. The latter two are particularly relevant given the wicked problems or polarities inherent in complex environments, which can never be totally resolved. This also involves challenging participants to check their assumptions and consider alternate or modified paradigms, even if they choose not to adopt them. Facilitators should also discuss the relevance to participants of program content through five leadership lenses: self, direct reports, peers (leading “beside”), senior leaders (leading “up”), and external colleagues (leading “beyond”).

#### 3.9.4. Feedback and Evaluation


**D10: (GS) Ensure participants receive formal feedback from several sources during the program**


As mentioned regarding RLCs and ELCs, receiving performance feedback effectively is key to enhancing self-awareness and to maximizing learning and development [7,19,35,65,82,130,229,230], since without such feedback, outcomes appear to be comparatively reduced [113,130]. Interim feedback has also been shown to decrease participants’ stress associated with challenging assignments, as well as increasing the likelihood that outcomes will be achieved successfully [75]. Feedback should come from multiple sources, including peers, faculty, experts, and workplace colleagues [60,136,161]. Feedback should be timely, constructive, and actionable [7,231,232], and based on credible pre-defined criteria and standards, such as those governing military After-Action Reviews following simulations [53]. These debriefs typically begin with lead participant self-ratings, followed by those of the peers involved and expert observers [7,44,224,225,226,227,228]. This variety highlights the degree of congruence among raters, which enables all participants to learn about effective performance, and about receiving and giving feedback, the latter of which is also a key leadership capability [225,233,234]. Feedback should also be gathered in reference to relevant participant goals, such as the capabilities they are targeting for development, and to projects, including participant performance, as well as the effectiveness (or not) of certain strategies. Following this process, the lead participant can summarize important lessons to consolidate learning.


**D11. Routinely provide time for participants to self-evaluate their progress, and have them share their results with their accountability teams and adjust their goals accordingly**


Similarly, participants should be given time regularly to reflect on their experience, feedback, and learning, as well as their relevance to the work context and to their LDPs [75,226,239]. These sentiments can be recorded in a journal, an exercise that serves as an additional developmental, heuristic tool [198,240]. When participants share their insights from reflection and results with their accountability teams, the ensuing conversation typically inspires further thoughts and support. Gauging progress during the program can signal when goals should be extended because headway is being made, or when extra support or adjustments are needed because difficulties are being encountered [65,130,145,258]. This adaptation can augment application outcomes [93], particularly with ongoing support from program personnel [65,93,95].


**D12. Collect informal feedback regularly from participants regarding effectiveness and proposed improvements and adapt the delivery accordingly**


Facilitators should routinely collect informal and anonymous feedback from participants to optimize program effectiveness. Data can include participants’ perspectives on delivery aspects that contribute effectively to achieving their goals and should be continued, which aspects they find ineffective and should be modified or dropped, and whether additional approaches could be introduced to improve results. These responses should be analyzed, summarized, actioned, and communicated speedily in an attempt to improve success, demonstrate a commitment to participants to incorporate their feedback in real time, and model the process [7,11,125,130,268].


**D13. (GS) Collect formal anonymous feedback regularly from participants on the program, communicate the results to relevant affected parties, adjust accordingly, and communicate adjustments to participants**


Concomitant with the previous strategy, providers should formally evaluate programs and their components to assess quality, enhance program credibility and satisfy accreditation requirements, and inform during-program modifications to optimize the experience [19,50,130,269]. It is helpful if anonymous surveys are administered in class to maximize the response rate and should involve quantitative ratings of sessions based on their stated goals, of faculty, and of activities linked to outcomes, as well as open-ended questions regarding effective elements and suggestions for improvement [27]. Select results may be communicated to affected parties, including faculty, which augments accountability for each person’s role in achieving their desired outcomes.

#### 3.9.5. Curriculum


**D14. (GS) Provide tools and resources (frameworks, checklists, models, technologies, services, etc.) that participants can sample in class and apply to their work**


Despite statistician George Box’s claim that all models are wrong, but some are useful [270], providing participants with credible theoretical and practical tools and resources is important to enhance content validity, focus discussion, and guide application and assessment. For example, following a program session on crisis leadership, it would likely be beneficial for participants to use a validated checklist of protocols as the basis for a workplace readiness assessment or group self-evaluation following an incident, rather than using notes taken during a presentation. Participants can also share these tangible takeaways with non-program colleagues. Resources can include those to which participants have access within or through their organization, as well as those publicly available. Along with theoretical models and frameworks, practical resources can include available technologies, websites, and services. Participants are more likely to apply tools at work and be confident doing so if they gain experience using them in class.


**D15. Engage with the dynamic and contextual nature of leadership and address emerging important internal and external issues as they arise**


Given the dynamic and contextual nature of leadership, along with the theoretical applications of preferred approaches to suit different circumstances, faculty should include emerging issues from participants’ professional context as real-life, real-time case studies. Internal matters can include those participants raise, whether interpersonal, process-related, organizational cultural challenges-related, or major initiative-related, and external ones can pertain to new government legislation that will impact the organization directly. This process enhances the relevance of course content to participants’ dynamic professional context and demonstrates how they might apply it to work.


**D16. Gather input from participants and other affected parties on emerging priority topics and include it in the program as it develops**


Following from the design strategy to reserve some program time for emerging priorities, faculty should consult participants and other affected parties regarding priority topics and issues in the internal or external context. A select few should be included in the program, which can manifest in many forms, such as in discussions or designated sessions, such as master classes.

#### 3.9.6. Direct Transfer Strategies


**D17. Remind participants of the end-of-program culminating activity so they focus and prepare**


While regularly highlighting key program lessons and outstanding questions, facilitators should remind participants of the culminating activity (C3 below) and invite them to consider what they have found most valuable in a given session, further contrasted to previous insights, along with corresponding action items. Along with preparing them for their final presentations, these reflections help focus and consolidate session learning and prompt them to review takeaways from previous content.

### 3.10. Program Conclusion Leadership Development Strategies (“C”, n = 11)

#### 3.10.1. Design


**C1. In class, have participants reflect and provide feedback on each component of the program specifically to identify key learnings and ways to optimize, partly to prepare for the culminating activity (C3)**


Faculty should reserve program time during the final session for participants to provide feedback on the effectiveness of all aspects of the program curriculum. This involves providing copies of the goals, topics, faculty, speakers, and activities for each component and having participants reflect on key learnings for each part, as well as which were particularly effective or not in contributing to the achievement of outcomes. This information is helpful feedback for quality improvement; and additionally, the reflective and discursive process is equally intended to enable participants to consolidate their learning and prepare for the culminating activity.


**C2. Celebrate participants through a formal event at the conclusion and invite senior leaders, supervisors, and sponsors, perhaps with past graduates, to cultivate a community**


Having a formal graduation-like ceremony can be helpful in validating the investments all personnel have made, celebrating achievements and impact, and publicly committing to further actions. Having organizational senior leaders and sponsors attend, along with past program graduates, heightens the perceived importance of the event and program and enables participants to express gratitude for their support. Involving past graduates can grow and sustain a leadership community, which has positive organizational cultural benefits.

#### 3.10.2. Direct Application Strategies


**C3. Have a culminating activity whereby participants present their key learnings, program impact, and committed action items during the celebration event**


The culminating activity can involve participants making brief presentations of their most valuable lessons, key results and impact, including highlights from their projects, and further actions they commit to take. This exercise helps focus and consolidate participants’ learning and enhance their accountability during and following the program, particularly when they share their presentations with their accountability teams. Including this as part of the celebration event enables senior leaders to hear the testimonials, which provides preliminary justification of their investment through anecdotal examples of the program impact and ROI.


**C4. Remind personnel of the post-program assessments**


At the end of programs, to increase participant commitment to achieving results and to augment the outcomes themselves, it is helpful to remind participants and those assessing their performance or receiving results sometime after the program of their process.

#### 3.10.3. Design


**C5. Extend development by having each participant update their LDP with post-program career and development goals and plans, aligned to talent optimization strategies**


It is important to establish individual and organizational next steps to extend the impact of interventions. Having participants set formal career and development post-program goals and feedback systems can improve outcomes [7,22,50,223,230,271], avoid relapse to previous behaviors [272], and increase participants’ intrinsic motivation [203,273]. Meta-analytic data suggest that this process is maximized when assessments from development interventions are integrated into their LDPs, as part of their career pathways, and into the organization’s regular performance appraisal and reward systems [12,50,69,130,146].

Organizations should also identify further formal and informal developmental opportunities for graduates [11,34,63,115,122,145,274], which could include specifically designated next steps, such as a subsequent, more advanced educational course, or pursuing a formal credential [7,128,275]. It is helpful to present a menu of available options, such as MSF, coaching, and structured, on-the-job learning, such as action learning projects, stretch assignments, job shadowing, or job rotations [34,44,49,82,122,129,132,272,276].

In any case, further options should align to talent optimization strategies for each participant and the organization overall.

#### 3.10.4. Evaluation


**C6. (GS) Formally evaluate the program overall based on its goals, as well as its individual components and their link to outcomes**


Evaluation at the conclusion of a program can gauge its immediate effects and is useful for optimizing the experience over time (quality control). Many organizations neglect program evaluation altogether [53], perhaps because success is assumed [50,277], or due to fears of budgetary cuts or professional discredit if unfavorable responses are received [278,279]. Evaluation, however, is essential, and must go beyond participant satisfaction [50,74,130,145], since doing so can highlight aspects of interventions that were beneficial or not in facilitating the achievement of goals and desired outcomes, which can enable providers to modify their curriculum and practices for the next iteration [7,11,80,81,125,130,145,268]. It is also important to inquire about barriers that participants encountered so that these can be addressed for future programs, which was one of the key recommendations in the Yale School of Management leadership review [220]. Post-program surveys should gather anonymous quantitative ratings of the extent to which the program met each of its stated goals, as well as qualitative feedback on unexpected outcomes, elaborations on the numerical ratings, and open-ended suggestions for improvement [27].


**C7. (GS) Evaluate participants’ progress in relation to the desired outcomes and hold them accountable, recognizing successes and supporting improvement**


Along with evaluating the program, it is important to assess participants’ desired outcomes in comparison to data collected at baseline and at various stages during the program [27]. This involves their individual desired outcomes, as well as organizational data and those related to their impact projects, contrasted to the control group, when applicable. Evaluation is essential to guide and enhance participants’ ongoing leadership development, particularly when they are held accountable for their desired outcomes [7,44,49,50,82,130,131,198,203,244,280,281,282] and share results with their accountability teams. One example of this is described in the study by Culpin et al. (2014) [124], in which managing directors emailed participants directly after a leadership intervention asking, “What have you done as a consequence of your participation in the training?” Post-program evaluation also provides a new baseline of individual and organizational capacity and outcomes, which can serve as a benchmark for further development [7,201].

Organizations can extend successfully achieved goals and offer corresponding rewards. Intrinsic motivation, such as pride in one’s work and achievements, has been found to be more influential on the retention of development learning than extrinsic factors, such as pay increases and promotions [203,273]. Intrinsic rewards can be enhanced through communicating results internally, including to participants’ supervisors and sponsors.

Conversely, it is also important to support underperformance through further development, communities of practice, coaching, buddy systems, and/or mentoring as an ongoing organizational commitment to, and investment in, its people [24,49,69,82,125,126,129,130].


**C8. (GS) Calculate the ROI of the program and communicate the results selectively**


Despite potential intangible benefits of leadership development, there is increasing pressure to attach monetary values to its reported outcomes by calculating the ROI [50,130]. This process helps demonstrate the strategic value of leadership development and can generate further support for future interventions [50,130].

Ensuring that economic outcomes are included in the desired outcomes for participant impact projects and in the evaluation framework, along with the increase in motivation caused by awareness that ROI will be calculated, can contribute to this process [50].

Program ROI can be calculated in several ways. Approaches typically involve comparing the cost of the program, including the opportunity cost of participants being off work, to the net benefits and financial outcomes achieved, such as increased revenue or productivity [130]. For example, Cascio and Boudreau (2011) calculated the mean effect of outcomes and the duration of that effect, multiplied by the number of participants, and compared this result to the performance of an in-house, non-program control group [283]. When monetized performance data were not available, the financial ROI was calculated by multiplying the increased participant performance compared to that of the control group, using a percentage of the participants’ annual salary. Jeon et al. (2013) describe monetizing outcomes by enumerated the costs saved by reduced absenteeism and turnover [121]. Avolio et al. (2010) also propose calculations for longer programs and the different levels of seniority of participants [50]. Even when program ROI calculations are negative, organizations may still choose to invest in leadership development because they either value ongoing development and its effect on recruiting, organizational or community image or branding, and staff retention, or their priority targets are not easily monetized [50]. This suggests that ROI calculations are only one factor to consider regarding subsequent investment in leadership development [50,284].

#### 3.10.5. Organizational Support


**C9. Communicate the results of programs to affected parties within the organization to celebrate successes, justify the investment, and provide support for future programs**


Communicating the results of leadership development is an opportunity to recognize the efforts of designers, faculty, and participants, to demonstrate its value and impact, and to increase credibility and support for future interventions [7,11,50,130]. These results can derive from formal assessments, project results, and anecdotal accounts of impact, including those offered during the culminating activity. Lauding participants’ achievements and results can also enhance their intrinsic motivation [203,273] and can help organizations identify internal champions for deeper leadership integration. Results can be shared through internal media communications, as well as directly to senior leaders and to participants’ supervisors and sponsors.


**C10. (GS) Ensure that there is adequate organizational support and resources to continue to facilitate further development**


Despite the many factors that support can contribute to making leadership development a transformational experience for individuals, it is what happens following programs that typically determines the extent to which learning is successfully applied and outcomes are achieved. The frequency of diminishing outcomes after interventions is largely attributable to participants reverting back to previous behaviors [63,285]. For example, Santos and Stuart (2003) found that 64 per cent of 167 financial service managers defaulted to their original work styles after development programs [135]. Regression to a previous state can also result from a lack of supervisor support or an organizational culture that inhibits efforts to innovate and experiment [49,52,184,286].

A clear LDP that extends beyond individual interventions, along with organization support, can counteract this trend [134]. This support begins by ensuring participants have opportunities to apply their learning after the intervention, particularly if this was not embedded [24,50]. A fertile workplace culture that facilitates leadership development means that participants’ colleagues must be willing to change too [49]. Proximity to colleagues who are also applying leadership learning increases motivation to operationalize new knowledge and skills. This fosters an open and psychologically safe environment with a common language to discuss leadership ideas, and offers peer support and mentoring [49], as well as augmenting the social contract effect. This support should be reinforced by recognition and reward systems, as well as by the provision of requisite resources to continue to expand capacity and efficacy [69]. The latter refers to providing further formal and informal developmental options for graduates [11,34,63,115,122,145,274], since participants become discouraged if there are inadequate follow-up opportunities and support [115,134,287].


**C11. Review program results in the context of the Leadership Integration Blueprint and Roadmap, develop a community and culture of leadership, and build internal capacity**


As described in F8 above, the impact of leadership development is maximized when it is fully integrated with talent optimization, embedded as an expectation of all people in the organizational culture, and seen as an ongoing strategic investment in individual and organizational capacity and performance, rather than as one-off events [7,130]. The Leadership Integration Blueprint and Roadmap, that is, the strategy moving forward, are key components of this, which can provide additional context within which to review post-program evaluations and identify designated further development opportunities for graduates. Organizations should also consider how to engage program graduates in cultivating an internal leadership community and how that group may be encouraged to entrench leadership more deeply in the fabric of the organization so that it becomes part of the “DNA” of all staff, regardless of their role or level, as in an ALO. Finally, it is important to build an internal capacity to develop leaders and leadership and to encourage and support innovation. This can involve holding supervisors accountable for developing their direct reports, with the support of HR/OD professionals, and creating structured goal setting and on-the-job development initiatives with this aim.

### 3.11. After-Leadership Development Program Strategies (“A”, n = 5)

#### 3.11.1. Evaluation


**A1. (GS) Evaluate the program, its components, and participant outcomes again from the perspective of sustained learning linked to application and results**


Data collected six to nine months after a program (post-post) can provide evidence of the extent to which improved outcomes have been sustained, as well as prompting participants to reflect again on their learning [27,53]. This is important since achieving outcomes can take time, resulting in a false negative result if restricted to end-of-program [19,247,248,249]. Conversely, participants’ initial post-training confidence in their abilities, motivation, and self-efficacy can decline after attempting to apply learning to the workplace [7,12,127,259]. For example, the study by Arthur Jr. et al. (2003) found a 90 per cent atrophy in participants’ skills a year after a training intervention [28]. Furthermore, entertaining speakers and high-energy sessions typically earn top ratings immediately after; however, this satisfaction does not always translate most effectively (or at all) to workplace application and to improved performance, when assessed sometime after. The inverse can also be true. Post-program ratings also provide another opportunity for participants to describe unanticipated outcomes.

#### 3.11.2. Direct Application Strategies


**A2. Providers remind participants, their supervisors, and their sponsors in advance of the post-program ratings of participant outcomes**


Response rates and performance outcomes increase when providers inform participants and their supervisors and sponsors prior to sending post-program evaluations and their results. For example, when participants are reminded that in two weeks, their supervisor will rate their performance and/or will be notified of participants’ outcomes, the participants’ motivation to demonstrate results, including by initiating workplace application, increases. This social accountability strategy is also an opportunity for participants to illustrate how they have put the investment to use to benefit the organization.


**A3. Have participants report results of their progress regarding post-program goals**


In addition to further reflections on program impact and application strategies, providers should also invite participants to share outcomes related to their post-program goals.

#### 3.11.3. Design


**A4. (GS) Consolidate the various forms of feedback and revise the program based on the feedback and evolving organizational needs**


Providers should ensure that along with modifications and improvements based on feedback throughout the program, they organize a formal debrief of all program evaluations with key affected parties selectively. Discussion should concern feedback-inspired suggested program revisions, as well as revisions to accommodate evolving organizational needs or a different participant group. Seizing this opportunity primes providers for the needs, gaps, and opportunities analysis of the next one(s).


**A5. Sustain the program community of graduates, including through refreshers and involvement in future iterations of the program (e.g., as speakers, mentors, etc.)**


Building on the benefits of networking described earlier, involving program graduates in cultivating or broadening a community of internal leaders who can champion further leadership development and integration is an important further step.

## 4. Discussion

The leadership development literature tends to focus primarily on interventions themselves, often with little or no mention of the application of learning strategies or of leadership development and integration across an organization. While there is increasing evidence that leadership development can be effective in facilitating desired outcomes at different levels, if theory- and evidence-informed approaches and transfer considerations are overlooked, even outstanding programs that incorporate gold-standard elements can underperform or fail. This possibility, along with dated theoretical models and widespread uncertainty regarding reliable evidence, has led to calls for a framework of theory- and evidence-informed approaches to leadership development and strategies that can be implemented before, during, and after programs to maximize their impact and return on investment (ROI). In the absence of such a framework, the objective of this article was to create one, called the “Optimizing System”.

To identify strategies that are empirically proven to facilitate the outcomes and ROI of leadership development, five data sources were analyzed: the forthcoming companion article by Geerts (2024), which establishes the theoretical and empirical foundations for the framework [88], and three systematic literature reviews on leadership development for professionals, with a combined data set of 172 unique empirical studies from 2000 to 2020, including 30 high-quality studies. The fifth was a literature review specifically on the application of learning. This article builds on previous research [7,19,32,34] by combining robust theory, gold-standard empirical evidence, meta-analytic data, and leading practice examples from multiple sources to create a novel theory- and evidence-informed framework that can assist practitioners in designing, delivering, evaluating, refining, and supporting leadership interventions to maximize their effectiveness [7]. The Optimizing System and the other included innovative models and frameworks also have the potential to influence further research in this burgeoning field.

### 4.1. The Optimizing System Framework

The Optimizing System includes unique theoretical models and elements of gold-standard program designs and strategies for maximizing impact. Following a foundational set, the strategies are organized according to when they can be implemented: before, during, at the conclusion of, and sometime after leadership development programs. They are subdivided into major categories: program design, delivery, personnel (participants and faculty), direct application strategies, evaluation, and organizational culture and support (Table 3).

### 4.2. The Scarcity of Evidence-Based Strategies in Practice

Overall, despite the gold-standard evidence informing these strategies, they are seldom incorporated in programs described in academic publications [27,53], which indicates a significant research/practice divide [288,289]. This inspires two questions. First, is this pool of interventions representative of leadership development practice somewhat generally, or might many organizations be implementing the approaches and strategies regularly but not publishing their results?

Second, why might these strategies not be being implemented consistently? Along with the aforementioned ambiguity regarding which approaches are grounded in the best available evidence [25,27], some authors suggest that organizations may lack the expertise or capacity to research optimal strategies. Others suggest that they may question their value [7] and/or feasibility in terms of the time and cost of incorporating them [85,130], as well as wishing to avoid evaluation fatigue or overburdening staff who would be tasked with analyzing feedback [85,130]. The former justification is more likely if past evaluations did not result in demonstrable changes made in their response [130]. Similarly, organizations may neglect calculating the ROI because they consider it too complex, unreliable [50,130], or politically risky, especially if there are concerns that negative responses may result in budgetary cuts or criticism of those who designed or advocated for them [278,279]. Regardless of the reason, the consequences of these strategies being shortchanged or excluded are that participant outcomes and program impact suffer as a result.

## 5. Implications for Further Research and Practice

Based on the findings of this research, several opportunities for further research and practice are proposed. The first concerns expanding the pool of gold-standard elements of leadership development (Table 1) and investigating aspects of leadership development for which there is not yet definitive evidence, such as which combinations of developmental activities are most effective at specific stages of people’s careers. That is, for whom, to what extent, and in what circumstances are certain components more effective than others.

Second, validating the Optimizing System, as well as the models and frameworks undergirding it, with practitioner and academic experts would be advantageous [288,289,290]. Along with potentially informing whether any elements of the models should be modified, added or cut, the validation process could also reveal the extent to which they and the strategies are generic versus contextual in terms of sector, profession, level of seniority, and other variables.

Third, it would be helpful to directly compare programs that implement the models and Optimizing System to similar interventions that do not, which could indicate the extent to which the framework can have an impact on outcomes.

Fourth, identifying and propagating leading-practice examples of how organizations can effectively implement the models and frameworks into the design, delivery, evaluation, and support of leadership development, such as with the Inspire Nursing Leadership Program [105], are important research priorities. This would also offer further examples of outcomes at various levels, particularly beyond that of the participant, which can be successfully achieved through leadership development. Extending the scope of leadership development beyond individuals to that of the team, organization, beneficiaries (clients), community, region/nation, and an international/global audience, as well as targeting core outcomes of environmental; equity, diversity, inclusion, and accessibility (EDIA); and economic value, can require a paradigm shift, which will likely take time to be adopted on a broader scale.

Fifth, it is important to investigate how the models and framework, whether as presented here or with modifications, could be optimally applied to different geographical locations and contexts, such as different stages of a crisis, and in reference to different leadership models, such as distributed, relational, transformational, authentic, servant, adaptive, and True North leadership.

Sixth, an ongoing priority is considering how new technologies, such as simulators and Artificial Intelligence (AI), might be incorporated into development programs effectively [7]. There is some evidence that principles of effective design outweigh the specific media of delivery [291], while others suggest that technology-based training is preferred [7].

Seventh, it would be useful to investigate how aspects of talent optimization can be linked explicitly and symbiotically to leadership development programs and evidence-based strategies for deepening leadership integration across organizations and communities, toward the concept of A Leadership Organization (ALO). This research could illuminate comprehensive examples of Leadership Integration Blueprints and Roadmaps that other organizations might follow.

Eighth, although rarely mentioned in the academic literature, it would be interesting to consider the role that leadership development could play in reaching beyond training centers and organizations to positively impact communities and beyond for social change. The possibility of this scope echoes the contention of educational theorists, such as Freire (2007) [95], Palmer (1998) [292], and Merriam and Brockett (1997) [293], who advocate that this is an essential function of education to facilitate. Given the influence that leaders, leadership, and organizations can have on communities and, collectively, on government and policy, as well as the increasing need for cross-sector collaboration, development programs that prepare people to influence on this scale have the potential to be highly impactful [294]. Generating evidence of this, particularly including economic outcomes, may encourage governments to invest in leadership development for multiple organizations, groups, and communities. Optimal strategies to maximize the impact of this scope of leadership development are worthy of further investigation, as are leading-practice examples of successful programs.

A final point of discussion concerns how the Optimizing System and its supporting models might reflect and inform effective leadership generally. For example, involving affected parties’ input in important strategic decisions, reinforcing accountability for performance outcomes, and providing organizational support and resources are key leadership practices. Similarly, this article has stressed the critical relationship among leadership development, talent optimization, organizational priorities and strategy, True North elements, and organizational culture, the interconnectivity of which is often overlooked.

## 6. Limitations

While this article provides several potential contributions to scholarship and practice, some limitations should be considered. First, the literature searches were limited to English-language publications, most of which concerned leadership theory and interventions in Western countries. Consequently, the representative nature of leadership development globally is unclear. Similarly, unpublished studies were excluded, limiting the knowledge base and exposing potential concerns of publication bias in favor of interventions that reported positive outcomes; however, this choice was made to prioritize the most credible available evidence and several included articles described failed outcomes. Third, in multi-component interventions, it can be challenging to assess the impact of individual aspects or combinations. Finally, although the strategies in the Optimizing System framework and supporting models are informed by robust theory and evidence, further validation and empirical support would be beneficial. That said, the framework and models presented here have the potential to credibly guide the design, delivery, evaluation, and support of leadership development to maximize its impact and ROI.

## 7. Conclusions

The escalating investments in leadership development and the concomitant expectations that it will generate positive outcomes and a demonstrable ROI heighten the urgency for evidence and tools to optimize its impact on organizations and beyond. The greater the stakes, such as sectors where people’s lives or well-being depend on effective leadership, the more vital it is for development to be designed based on empirically informed strategies. There is ample evidence that leadership development can facilitate the achievement of outcomes at the individual, team, and organizational levels; yet, many gold-standard design components are seldom utilized, and they tend to be divorced from discussions of the application of learning strategies, the importance of organizational culture, and the value of integrating leadership and development system-wide. These disconnects and the failure to address “the transfer problem” can have numerous deleterious consequences [130].

The framework presented here is informed by robust theory, gold-standard evidence, meta-analytic data, and leading-practice examples from individual empirical studies with the intention of guiding the design, delivery, evaluation, and support of leadership development to maximize results and ROI. Innovative supporting models are also presented, many of which challenge traditional paradigms. These models include principles of leadership development, progressive levels of mastery, an outcomes-based design, levels of program outcomes, categories of development activities, a program evaluation framework, talent optimization functions, and the concept of A Leadership Organization.

Collectively, these models require paradigm shifts to extend the scope beyond single development interventions for positional leaders that target individual-level outcomes, to viewing leadership as a vital, comprehensive, interconnected component of talent optimization and a catalyst of leadership integration across organizations and communities that can facilitate outcomes at multiple levels and even social change.

The Optimizing System framework presented here also includes an evidence-informed emphasis on expertise and equity, diversity, inclusion, and accessibility (EDIA). This framework can potentially inform the design and support of individual leadership development programs, as well as organization-, system-, or community-wide leadership development and integration.

Given that the impact of leadership development is influenced significantly by how programs are designed, delivered, evaluated, and supported, the hope is that the Optimizing System framework can assist governments, policymakers, providers, those funding development, and other affected parties in offering development that produces the desired results and maximizes the ROI. The further hope is that this ROI will reach outward, beyond individual organizations, to significantly benefit the communities and regions in which these organizations have made their home.

## Figures and Tables

**Table 1 behavsci-14-00955-t001:** Gold-standard elements of program design, delivery, and evaluation.

Category	Element	Element Details
Design	Conduct a pre-program needs and gaps analysis	
	Apply an outcomes-based design approach	Please see Table 4 for an example.
	Define explicit program goals	
	Have participants select personalized goals	
	Incorporate the principles of adult learning [93]	(Pre-program): Motivation to learn; 1. Self-directed; 2. Participants’ experience as the basis; 3. Content that is practical and relevant to participants; 4. Outcomes-based
	Incorporate a leadership capability framework	For example, the LEADS in a Caring Environment framework
	Embed an application of the learning/training transfer strategy	
	Calculate the return on investment (ROI)	
Personnel: faculty	Mixed	Internal/external to the organization; and experts/practitioners
Design: developmental activities	Individual	Multi-source feedback (MSF) and 360-assessments, psychometrics, goal setting, Leadership Development Plans (LDPs), certifications
	Educational	Lectures, webinars, small-group discussions, guest speakers, case study analyses, site visits, and assignments
	Experiential	Role play, simulations, training, impact projects, application exercises, internships, job shadowing, and presentations
	Relational	Individual and peer coaching, mentoring, networking, and engagement in a Community for Practice
	Resources	Readings, reflective journals, toolkits
Outcomes	Desired outcomes based on the Kirkpatrick (2006) framework [96] levels: satisfaction (1), attitudes and perceptions (2a), knowledge and skills (2b), subjective behavior change (3a), objective behavior change (3b), organizational change (4a), and benefit to patients (4b)	For a revised framework, please see Table 5.
Evaluation	*Focus* of evaluation: both the program and participant outcomes	
	*Type of data* collected: both quantitative and qualitative	
	*Type of data* collected: both objective and subjective	
	*Raters*: multiple	Self, peer, program faculty, workplace supervisor
	*Control* group	
	*When* data were collected: pre/baseline, post, and post-post	

**Table 2 behavsci-14-00955-t002:** Principles of leadership development [88].

No.	Principle of Leadership Development
Pre-program	Motivated and invested personnel with shared accountability
1.	Timely, customized, co-created, contextual, and aligned with the organizational needs and opportunities, True North elements, including the strategic plan/priorities, the talent optimization strategy, and the organizational context
2.	Integrated, coordinated, and embedded in the organizational culture, system-wide, including in all aspects of talent optimization
3.	Evidence-informed (design), including regarding transfer to work
4.	Experiential and application- and outcomes-based
5.	Aligned with the principles of adult and professional learning, including being multi-faceted with optionality
6.	Capacity-growing through increased self-awareness, efficacy, and adaptability
7.	Evaluated (program and outcomes) and evolving (during and after)
8.	Explicit about prioritizing expertise and equity, diversity, inclusion, and accessibility (EDIA)
9.	Enduring (individual) and sustainable (organizational)
10.	Relational and community-centered

**Table 3 behavsci-14-00955-t003:** The Optimizing System.

Foundations for Leadership Development Programs (“F”, *n* = 9)		
Type	No.	Strategy
Foundational models	F1 *	Have a shared leadership model and capability framework as the common conceptualization and language of leadership
Organizational culture	F2 *	Earn executive support for leadership development as a key strategic enabler and investment in their people
	F3 *	Embed leadership development as a key component of talent optimization, aligned with the overall organizational purpose, values, vision, mission, and strategy (True North)
	F4 *	Provide funding and/or resources and protected time for leadership development
	F5	Ensure there is a safe culture of learning and leadership within the program and in the organization
Design	F6	Ideally, have developed a comprehensive ecosystem of leadership development interventions, experiences, and resources available to staff at all levels, aligned to career pathways
Leadership integration	F7	Ideally, have distributed leadership organization-wide, fully integrated into talent optimization, embedded in the organizational culture, and is an accountability expectation of all staff to develop and support the development of others
	F8	Ideally, synthesize graphically the various forms of leadership development and integration, as well as their interconnectivity to each other and to career pathways for all people, in a blueprint
Organizational support	F9	Develop a communications strategy for the program to relay progress and celebrate achievements
**Pre-Leadership Development Strategies (“P”, *n* = 23)**		
**Type**	**No.**	**Strategy**
Context and engagement	P1 *	Conduct a needs, gaps, opportunities, and strategic priorities analysis involving key affected parties (stakeholders) to inform program goals and to generate engagement and support
Design	P2 *	Determine the scale and scope of the program, with a preliminary consideration of return on investment (ROI)
	P3 *	Apply a robust outcomes-based design strategy, beginning with selecting explicit program goals
	P4 *	Select ensuing desired outcomes, including level of mastery, at various levels
	P5	Ensure that diversity, equity, inclusion, and accessibility (EDIA) are prioritized in the selection of participants, faculty, speakers, and content
Participants	P6 *	Select participants intentionally to address organizational needs and priorities
	P7 *	Address participants’ motivation to learn and ensure that they can commit fully to the program with supervisor and sponsor support
	P8 *	Have participants create a Leadership Development Plan (LDP) that includes personalized goals and desired outcomes aligned to their organizational plans and career pathways
	P9 *	Ideally, align programs to internal and/or external professional requirements (e.g., Maintenance of Certification), distinctions, or credentials
Design	P10 *	Select evidence-based program details (content, faculty, developmental activities, resources, structure, etc.) intentionally to achieve identified targets, customized for the participant group
Faculty and guest speakers	P11 *	For faculty and guest presenters, prioritize diverse, mixed teams (internal/external and experts/practitioners)
Design: curriculum	P12 *	Customize the curriculum according to pre-program engagement and the participant group, and embed the leadership model and capability framework
	P13	Reserve in-program time for emerging issues/topics
Design: developmental activities	P14 *	Select developmental activities according to their intended impact on desired outcomes and offer a variety
	P15 *	Embed activities in reflective or experiential learning cycles
Direct transfer strategies	P16 *	Ensure that application/transfer is included explicitly throughout
Design: structure and formal	P17 *	Select the optimal program structure and format
Delivery	P18 *	Incorporate the principles of leadership development and adult learning
	P19	For programs involving mixed participant samples, include designation-specific cohorts or syndicate sessions
Evaluation	P20 *	Develop a robust evaluation framework for the program, including the ROI, and for participants, and consider a control group
	P21 *	Establish relevant baseline measures
Direct transfer strategies	P22 *	Conduct a barriers and enablers assessment and apply results to remove or circumvent obstacles and leverage enablers
	P23 *	Communicate explicitly the purpose, goals, content, outcomes, and evaluation framework of programs overall and individual components to establish a shared understanding and accountability among providers, faculty, participants, and other affected parties
**During-Leadership Development Strategies (“D”, *n* = 17)**		
**Type**	**No.**	**Strategy**
Direct transfer strategies	D1	Have participants select and engage accountability teams
Design	D2	Host a program launch and orientation event with senior leader support to level set and introduce personnel
	D3 *	Provide networking opportunities with program colleagues and faculty, internal senior leaders, guest speakers, past program graduates, etc.
	D4	Immerse in an internal Community for Practice (when available) to discuss, share resources, and connect more widely
Delivery	D5 *	Communicate to faculty and participants the purpose, relevance, rationale, and evaluation of each session, as well as the connection to other sessions and components
	D6 *	Maximize participant engagement and hold them accountable for attendance, in-class participation, and application/asynchronous exercises
	D7 *	Demonstrate a personal interest in all participants and their individual learning and development, as well as actively incorporating their expertise
	D8 *	Embed discussions of experiences and lessons from application exercises regularly to enhance and extend learning, as well as promoting accountability
	D9	Highlight emerging key themes, learnings, tensions, and ongoing questions throughout through different lenses
Feedback and evaluation	D10 *	Ensure all participants are provided with formal performance feedback from several sources during the program
	D11	Routinely provide time for participants to self-evaluate their progress with respect to their LDPs, and have them share their results with their accountability teams and adjust their goals accordingly
	D12	Collect informal feedback regularly from participants regarding effectiveness and proposed improvements and adapt the delivery accordingly
	D13 *	Collect formal anonymous feedback regularly from participants on the program, communicate the results to relevant affected parties, adjust accordingly, and communicate adjustments to participants
Design: curriculum	D14 *	Provide tools and resources (frameworks, checklists, models, technologies, services, etc.) that participants can sample in class and apply to their work
	D15	Engage with the dynamic and contextual nature of leadership and address emerging important internal and external issues as they arise
	D16	Gather input from participants and other affected parties on emerging priority topics and include them in the program as it develops
Direct transfer strategies	D17	Remind participants of the end-of-program culminating activity so they focus and prepare
**Program Conclusion Leadership Development Strategies (“C”, *n* = 11)**		
**Type**	**No.**	**Strategy**
Design	C1	In class, have participants reflect and provide feedback on each component of the program specifically to identify key learnings and ways to optimize, partly to prepare for the culminating activity
	C2	Celebrate participants through a formal event at the conclusion and invite senior leaders, supervisors, and sponsors, perhaps with past graduates, to cultivate a community
Direct application strategies	C3	Have a culminating activity whereby participants present their key learnings, program impact, and committed action items during the celebration event
	C4	Inform personnel of the post-program assessments
Design	C5	Extend development by having each participant update their LDP with post-program career and development goals and plans, aligned to talent optimization strategies
Evaluation	C6 *	Formally evaluate the program overall based on its goals, as well as its individual components and their link to outcomes
	C7 *	Evaluate participants’ progress in relation to the desired outcomes and hold them accountable, recognizing successes and supporting improvement
	C8 *	Calculate the ROI of the program and communicate the results selectively
Organizational support	C9	Communicate the results of programs to affected parties within the organization to celebrate successes, justify the investment, and provide support for future programs
	C10 *	Ensure that there is adequate organizational support and resources to continue to facilitate further development
	C11	Review program results in the context of the Leadership Integration Blueprint and Roadmap, develop a community and culture of leadership, and build internal capacity
**After-Leadership Development Program Strategies (“A”, *n* = 5)**		
**Type**	**No.**	**Strategy**
Evaluation	A1 *	Evaluate the program, its components, and participant outcomes again from the perspective of sustained learning linked to application and results
Direct application	A2	Providers remind participants, their supervisors, and their sponsors in advance of the post-program ratings of participant outcomes
	A3	Have participants report results of their progress regarding post-program goals
Design	A4 *	Formally review the various forms of feedback and revise the program based on the feedback and evolving organizational needs
	A5	Sustain the program community of graduates, including through refreshers and involvement in future iterations of the program (e.g., as speakers, mentors, etc.)

NB: “*” = based on gold-standard elements.

**Table 4 behavsci-14-00955-t004:** An outcomes-based design process [27].

Steps	Details
1.	Conduct a needs, gaps, opportunities, and priorities analysis and establish an empirical foundation
2.	Select explicit program goals customized for the purpose and participants and aligned to the organization’s True North elements
3.	Select ensuing desired outcomes, including level of mastery, at various levels
4.	Select participants intentionally to address the needs, gaps, opportunities, and priorities
5.	Select program details intentionally and incorporate evidence-based elements according to their suitability to achieve identified targets
6.	Develop a robust evaluation framework
7.	Embed an application of learning/training transfer strategy
8.	Consider calculating the return on investment (ROI)

**Table 5 behavsci-14-00955-t005:** A framework categorizing leadership development program outcomes [88].

Scope	Level	Details
**Individual**		
	1	Satisfaction
Participant	2a	Attitudes and perceptions
	2b	Increased knowledge, skills, and capabilities
	2c	Paradigm/mindset shift
	3s, 3o	Behavior change and performance improvement (subjective/objective)
**Beyond**		
Team	4s, 4o	Team impact (subjective/objective)
Organizational	5s, 5o	Organizational and staff impact (subjective/objective)
Beneficiaries	6s, 6o	Beneficiary impact (subjective/objective)
Community	7s, 7o	Community impact (subjective/objective)
Region/nation	8s, 8o	Regional/national impact (subjective/objective)
International/global	9s, 9o	International/global impact (subjective/objective)
**Core outcomes**		
	Env.s, Env.o	Environmental sustainability impact (subjective/objective)
	EDIAs, EDIAo	Equity, diversity, inclusion, and accessibility (EDIA) impact (subjective/objective)
	Ec.s, Ec.o	Economic impact (subjective/objective)

NB: “s” = subjective; “o” = objective.

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
