# Peer review of "Maximizing the Impact and ROI of Leadership Development: A Theory- and Evidence-Informed Framework"

_behavsci, 2024, doi:10.3390/bs14100955_

Round 1
Reviewer 1 Report
Comments and Suggestions for Authors
I have found the article very interesting, yet quite questionable. It contributes to the existing literature but it should be up to the editors to firstly decide whether it is suitable to be published in Behavioral Sciences in this form. It is definitely not a research article, it conceptualizes several existing issues and provides some essential advice for practice but I do not know if 50+ pages of theoreticizing is something the renowned "BS" journal wants to undersign...
My suggestion to the author is to publish this study as a short book rather than a research article as it cannot really be defined as such.
If there is a Special Issue of Behavioral Sciences which would welcome this paper, I recommend it to be published.
Author Response
Thank-you for your time, supportive comments, and feedback, the latter of which I have incorporated and will address point-for-point below. My responses are in bold.
Comments and Suggestions for Authors
I have found the article very interesting, yet quite questionable. It contributes to the existing literature but it should be up to the editors to firstly decide whether it is suitable to be published in Behavioral Sciences in this form. It is definitely not a research article, it conceptualizes several existing issues and provides some essential advice for practice but I do not know if 50+ pages of theoreticizing is something the renowned "BS" journal wants to undersign...
We have changed the type of article to “review”
My suggestion to the author is to publish this study as a short book rather than a research article as it cannot really be defined as such.
If there is a Special Issue of Behavioral Sciences which would welcome this paper, I recommend it to be published.
I agree the length is considerable and I have looked for ways to shorten it. Thank-you again for your endorsement; I appreciate it.
Reviewer 2 Report
Comments and Suggestions for Authors
Dear author,
Thank you for addressing a relevant and very interesting topic in leadership research.
Please find below the comments.
Introduction. The references are properly framed, and this section is very well articulated. However, it would be useful to mention what type of organizations are considered in the study or include more illustrative examples in different sectors (business, education…). The manuscript mentions healthcare settings but adding examples would strength the scope of the work. Regarding the subsection “Definitions”, a more narrative writing style is recommended and a brief paragraph justifying the selection of concepts.
Methodology. The methodology is clear, accurate and very well described. As a recommendation, some explanation about the inclusion and exclusion criteria used in the selection process would improve this section.
Results. The results are clearly explained and provide a coherent description for leadership research.
Discussion. The discussion is presented with soundness and rigour.
Implications – limitations. Overall, the implications for future research and limitations accurate and coherent. However, there is a lack of explanation or reference to the different leadership styles or models (shared, transformational…) and how they can generate different effects.
Conclusion. The conclusion is well addressed.
Hope the comments are helpful.
Thank you very much,
Author Response
Thank-you for your time, supportive comments, and feedback, the latter of which I have incorporated and will address point-for-point below. My responses are in bold.
Comments and Suggestions for Authors
Dear author,
Thank you for addressing a relevant and very interesting topic in leadership research.
Thank-you for your kind words.
Please find below the comments.
Introduction. The references are properly framed, and this section is very well articulated.
Thank-you.
However, it would be useful to mention what type of organizations are considered in the study or include more illustrative examples in different sectors (business, education…). The manuscript mentions healthcare settings but adding examples would strength the scope of the work.
Thank-you for this suggestion. In response, we have added in the methods section the following: “included 56 empirical studies from various sectors, including business and private sector, the military, public sector and government, manufacturing, engineering, healthcare, public health, and higher education.
Hopefully this will provide a sense of the breadth of sectors/domains involved. As for examples, we’ve been edited to crop, not add, sadly, but there is certainly an opportunity for follow-up articles describing examples of the framework in action.
Regarding the subsection “Definitions”, a more narrative writing style is recommended and a brief paragraph justifying the selection of concepts.
Excellent suggestion. As a result, we included the following:
Defining what we mean by “leadership”, “leadership development”, “integration”, etc. matters because the terms are used so commonly, yet inconsistently, and there is no collective set on which scholars and practitioners generally agree[1]. And yet, how one understands leadership, as distinct from “management” and “power”, for example, has significant implications for how one approaches research, practice, and development program design, delivery, and evaluation[1], [2]. Similarly, the predominant term for the enterprise, “leadership development”, nearly always refers to interventions about leadership for individuals (thus, leader development), often with minimal or no accountability-bound expectations of involving their colleagues. This situation rarely yields remarkable results[1], [2], further signalling the need for reliable research and guidance to maximize impact and ROI[1].
To clarify the nomenclature and theoretical underpinnings of the framework[1], the following definitions related to a) leadership, management, and power, b) development, and c) leadership integration are presented:To clarify the nomenclature and theoretical underpinnings of the framework[1], the following definitions related to leadership, development, and integration are presented:
Methodology. The methodology is clear, accurate and very well described. As a recommendation, some explanation about the inclusion and exclusion criteria used in the selection process would improve this section.
In response, we have added the following:
The reviews were informed by The Preferred Reporting Items for Systematic Reviews and Meta-Analyses (PRISMA)[1] and the Cochrane Review Handbook for Systematic Reviews of Interventions[1]. Studies identified in English language peer-reviewed academic journals were included[1] if they:
- focused on leadership education, development, or training interventions, programs, or individual developmental activities, such as coaching,
- featured adult professional participant samples, not school children, undergraduates, or pre-qualified trainees, such as military cadets or medical students, and
- evaluated the evaluating the effectiveness of the program/intervention, rather than simply presenting a model, theory, or a program that was not evaluated.
Programs featuring a single task, such as creating a business plan, capability, such as innovation, or interventions where leadership was only one of many learning outcomes, were not included[1].
Results. The results are clearly explained and provide a coherent description for leadership research.
Thank-you.
Discussion. The discussion is presented with soundness and rigor.
Thank-you.
Implications – limitations. Overall, the implications for future research and limitations accurate and coherent. However, there is a lack of explanation or reference to the different leadership styles or models (shared, transformational…) and how they can generate different effects.
Thank-you for this suggestion. In response, we have added the following to the discussion:
“Fifth, it would be important to investigate how the framework, whether as presented here or with modifications, could be optimally applied to different geographical locations, contexts, such as different stages of a crisis, and in reference to different leadership models, such as shared, transformational, authentic, relational, adaptive, and True North Leadership’
Conclusion. The conclusion is well addressed.
Thank-you.
Hope the comments are helpful.
Thank you very much,
Reviewer 3 Report
Comments and Suggestions for Authors
L 1-183: Excellent framing for the beginning. You do a great job of defining your terms and parameters as well as the inherency of the topic (and it's a very convincing argument). Glad this is being done.
184-224: Helpful way to show the data and categorization.
224-246: Really helpful chart to break down the elements in each category.
L641: Need to capitalize "intentionally."
761-777: This one really is key, and often overlooked.
850: "enrollment"
939: "Though" not thought
962: delete "are" from "are aware"
Reformat lines 1236-1237
thru 1467: This was very thorough, organized, and helpfully researched.
This was a really expansive, helpful summary of research and the essential elements of successful training. Thank you!
Author Response
Thank-you for your time, supportive comments, and feedback, the latter of which I have incorporated and will address point-for-point below. My responses are in bold.
Comments and Suggestions for Authors
L 1-183: Excellent framing for the beginning. You do a great job of defining your terms and parameters as well as the inherency of the topic (and it's a very convincing argument). Glad this is being done.
Thank-you for your kind words and support.
184-224: Helpful way to show the data and categorization.
Thank-you.
224-246: Really helpful chart to break down the elements in each category.
Thank-you.
L641: Need to capitalize "intentionally."
Fixed
761-777: This one really is key, and often overlooked.
Agreed!
850: "enrollment"
Fixed
939: "Though" not thought
Fixed
962: delete "are" from "are aware"
Fixed
Reformat lines 1236-1237
Fixed
thru 1467: This was very thorough, organized, and helpfully researched.
Thank-you!
This was a really expansive, helpful summary of research and the essential elements of successful training. Thank you!
Thank-you again!
Round 2
Reviewer 1 Report
Comments and Suggestions for Authors
Thank you for your response and related actions.